# Mononuclear diploid cardiomyocytes support neonatal mouse heart regeneration in response to paracrine IGF2 signaling

Hua Shen[1], Peiheng Gan[1,2,3], Kristy Wang[2], Ali Darehzereshki[4], Kai Wang[5], S Ram Kumar[6], Ching-Ling Lien[4], Michaela Patterson[7], Ge Tao[2], Henry M Sucov[2,3]*

[1]Department of Stem Cell Biology and Regenerative Medicine, University of Southern California Keck School of Medicine, Los Angeles, United States; [2]Department of Regenerative Medicine and Cell Biology, Medical University of South Carolina, Charleston, United States; [3]Department of Medicine Division of Cardiology, Medical University of South Carolina, Charleston, United States; [4]Saban Research Institute, Children's Hospital Los Angeles, Los Angeles, United States; [5]Department of Cardiovascular Surgery, the First Affiliated Hospital of Guangzhou Medical University, Guangzhou, China; [6]Department of Surgery, University of Southern California Keck School of Medicine, Los Angeles, United States; [7]Department of Cell Biology, Neurobiology and Anatomy, and Cardiovascular Center, Medical College of Wisconsin, Milwaukee, United States

**Abstract** Injury to the newborn mouse heart is efficiently regenerated, but this capacity is lost by one week after birth. We found that IGF2, an important mitogen in heart development, is required for neonatal heart regeneration. IGF2 originates from the endocardium/endothelium and is transduced in cardiomyocytes by the insulin receptor. Following injury on postnatal day 1, absence of IGF2 abolished injury-induced cell cycle entry during the early part of the first postnatal week. Consequently, regeneration failed despite the later presence of additional cell cycle-inducing activities 7 days following injury. Most cardiomyocytes transition from mononuclear diploid to polyploid during the first postnatal week. Regeneration was rescued in Igf2-deficient neonates in three different contexts that elevate the percentage of mononuclear diploid cardiomyocytes beyond postnatal day 7. Thus, IGF2 is a paracrine-acting mitogen for heart regeneration during the early postnatal period, and IGF2-deficiency unmasks the dependence of this process on proliferation-competent mononuclear diploid cardiomyocytes.

*For correspondence:
sucov@musc.edu

Competing interests: The authors declare that no competing interests exist.

## Introduction

In heart development, an initial pool of cardiomyocytes is derived by differentiation from early mesodermal progenitors. Once this early process has completed, in every later context in the embryo and throughout postnatal life, essentially all new cardiomyocytes are generated by proliferation of preexisting cardiomyocytes rather than by differentiation from a non-cardiomyocyte progenitor source (*Sereti et al., 2018*; *Li et al., 2019*). In mammals, the rate of embryonic cardiomyocyte proliferation is high as growth of the heart keeps pace with overall embryo growth, but then tapers down by the end of gestation (*Soonpaa et al., 1996*). In normal heart biology, cardiomyocyte proliferation is minimal for the remainder of life; the heart grows considerably in size postnatally, but this is by the hypertrophic enlargement of existing cardiomyocytes rather than by generation of new

cardiomyocytes. Nonetheless, in the context of an insufficient number of cardiomyocytes, the mammalian heart at late fetal and very early neonatal stages retains the ability to activate robust cardiomyocyte proliferation and to restore normal cardiomyocyte number. Although there have been some negative observations (*Andersen et al., 2016*), reactivation of cardiomyocyte proliferation to achieve efficient restoration of heart morphology has been observed in mouse genetic models that cause a non-lethal late-embryonic deficiency of cardiomyocytes (*Drenckhahn et al., 2008*; *Vincentz et al., 2017*), and in surgical or congenital injury models that cause a neonatal loss of ventricular myocardium in rats (*Robledo, 1956*), mice (*Sadek et al., 2014*), and swine (*Mahmoud and Porrello, 2018*), and likely also occurs in human (*Haubner et al., 2016*). In mice, the plasticity to reactivate efficient cardiomyocyte proliferation is mostly lost by postnatal day P7 (*Porrello et al., 2011*; *Alkass et al., 2015*), which is understood to mean that most ventricular cardiomyocytes have become postmitotic by this time. Consequently, heart injury in P7 pups or at any later time through adulthood is followed by insufficient cardiomyocyte proliferation and insufficient replacement of lost myocardium, i.e., by a failure to fully regenerate. In animals and humans, failure of regeneration after ventricular injury leads to a permanent diminishment of contractile function with possible further progression to heart failure.

A number of differences distinguish ventricular cardiomyocytes at the beginning and end of the first postnatal week, but one that may determine (rather than correlate with) proliferative ability is their total DNA content (*Gan et al., 2020*). In mice and perhaps most mammals, essentially all fetal and early neonatal cardiomyocytes have one diploid nucleus (as is the case for most cells). During the first postnatal week (in mice), cardiomyocytes undergo a last round of DNA replication followed by cell cycle arrest either before or after karyokinesis but without cytokinesis, resulting in mononuclear polyploid or multinuclear cardiomyocytes (*Soonpaa et al., 1996*; *Alkass et al., 2015*). Typically, only a few percent of ventricular cardiomyocytes thereafter and through the remainder of life are mononuclear diploid, which correlates with the very low level of cellular and functional regeneration usually seen after adult heart injury. Adult zebrafish retain a very high percentage of mononuclear diploid cardiomyocytes throughout adulthood (*Patterson et al., 2017*; *González-Rosa et al., 2018*), and heart regeneration is robust in this species (*Foglia and Poss, 2016*). Thus, from a comparison of multiple species and life stages, there is a precise correlation between the abundance of mononuclear diploid cardiomyocytes and competence to support cardiomyocyte proliferation for embryonic heart growth or post-injury heart regeneration. Recent experimental results in both mouse and zebrafish explicitly demonstrated that polyploid cardiomyocytes are compromised in proliferation and regeneration, and thereby confirmed the relationship between mononuclear diploid cardiomyocytes and adult heart regeneration (*Patterson et al., 2017*; *González-Rosa et al., 2018*). Of particular relevance to this paper, we showed (*Patterson et al., 2017*) that some inbred mouse strains have a much higher percentage of adult ventricular mononuclear diploid cardiomyocytes (e.g., 10% in strain A/J, compared to 2% in C57BL/6J), and that such strains have a correspondingly higher degree of both cellular and functional regeneration after adult heart injury.

Organ regeneration following injury or tissue loss is generally thought to recapitulate at least some features of embryonic growth, including the redeployment of signaling pathways that had been active earlier in fetal development. The early embryonic heart consists of cardiomyocytes surrounding an inner layer of endocardium (the endothelial lining of the heart). Around embryonic day E10 in mouse, the epicardium, the noncontractile mesothelium of the heart, migrates from a nearby source onto the outer heart surface. In our past work, we showed that the epicardium is the source of secreted mitogenic factors that promote cardiomyocyte proliferation (*Chen et al., 2002*), and then identified IGF2 (insulin-like growth factor 2) as the major epicardial mitogen responsible for this activity (*Li et al., 2011*; *Shen et al., 2015*). *Igf2* is expressed in the embryonic epicardium and endocardium (but not myocardium). Mouse embryos deficient in *Igf2* expression in all heart mesoderm or specifically in the epicardium lineage were substantially compromised in cardiomyocyte proliferation during the E10-E13 period, which was morphologically manifest in a hypoplastic ventricle. Notably for the present study, conditional *Igf2* mutation in the endocardium and endothelium had no phenotypic consequence in the embryonic heart (*Shen et al., 2015*). Embryos lacking the two signaling-competent IGF2 receptors INSR (insulin receptor) and IGF1R (insulin-like growth factor one receptor) in heart mesoderm or specifically in cardiomyocytes were also compromised in cardiomyocyte proliferation and ventricular morphology (*Li et al., 2011*), which is predominantly a reflection of the importance of IGF1R rather than INSR in the embryonic heart (*Wang et al., 2019*). Although the

midgestation hypoplastic ventricular phenotypes of conditional *Igf2* or IGF receptor mutants were obvious, these were not so severe as to cause embryo lethality. Cardiomyocyte proliferation in mutants then recovered to normal levels at E14 (*Li et al., 2011*), likely in response to other factors distributed by coronary circulation that begins at that time (*Cavallero et al., 2015*). Mutant pups were born in normal numbers and of normal size and health, and conditional *Igf2* mutants are viable and appear normal throughout a normal lifespan. Because conditional *Igf1r/Insr* mutants are also normal at birth, the other factors that restore cardiomyocyte proliferation in late gestation in *Igf2* mutants seemingly do not signal through IGF1R or INSR and thus are not likely to be IGF1 or insulin.

In this study, we address the reutilization of IGF signaling in neonatal heart regeneration. We find that IGF2 is a required factor for neonatal heart regeneration, although with mechanistic features that are distinct compared to its role in embryonic heart development. After neonatal heart injury, IGF2 is uniquely active during the early part of the first postnatal week when most cardiomyocytes are mononuclear and diploid. A later activity initiates cell cycle entry but is not able to support regeneration, at least under typical circumstances. Most significantly, several independent manipulations that elevate the percentage of mononuclear diploid cardiomyocytes allow rescue of regeneration in *Igf2*-deficient neonates.

## Results

### Endocardial IGF2 is required for neonatal mouse heart regeneration

In the embryonic heart, *Igf2* mRNA is expressed in the epicardium and endocardium, although not in the myocardium (*Li et al., 2011*). As noted above, genetic analysis demonstrated that only epicardial *Igf2* has a required role in heart development. We confirmed that the same expression pattern persists into the early postnatal period, although epicardial expression decreased noticeably from the middle of the first postnatal week and was essentially absent by postnatal day P7 (*Figure 1A,B*; *Figure 1—figure supplement 1A*). Endocardial *Igf2* expression at the RNA level was diminished quantitatively by P7, although was still clearly expressed. At no point was myocardial expression of *Igf2* detected.

Apical resection of the neonatal mouse heart at P1 is followed by robust cardiomyocyte proliferation and efficient regeneration (*Porrello et al., 2011*). We noted that apical resection did not affect *Igf2* expression, spatially or quantitatively (*Figure 1C–D*). Interestingly, the total amount of ventricular IGF2 protein was unchanged from birth through P7 and regardless of injury (*Figure 1E*; *Figure 1—figure supplement 1B*), despite the loss of epicardial *Igf2* expression and the diminished level of endocardial *Igf2* expression. The protein present at P7 may have been translated several days earlier, and may be stabilized and/or sequestered by binding proteins. In the uninjured heart, there is no apparent requirement for this protein in postnatal heart biology, as genetic *Igf2* manipulation with several *Cre* lines (*Shen et al., 2015*) had no recognizable postnatal consequence.

In our previous studies (*Li et al., 2011*; *Shen et al., 2015*), we used *Nkx2.5-Cre* to ablate *Igf2* function in all heart mesoderm. Because of its expression in early heart mesoderm lineage progenitors (*van Wijk and van den Hoff, 2010*), *Nkx2.5-Cre* drives efficient recombination in endocardium, epicardium, and myocardium, although at later stages, NKX2.5 protein is only expressed in cardiomyocytes. As noted above, *Nkx2.5-Cre/Igf2* mutants have a hypoplastic ventricle phenotype and reduced cardiomyocyte proliferation during embryonic days E10-13, although recover their level of cardiomyocyte proliferation at E14 and survive embryogenesis and have a normal adult lifespan. Other models with an early non-lethal cardiomyocyte deficiency have also demonstrated extension of cardiomyocyte proliferation in late gestation to fully reconstitute the cardiomyocyte population by birth (*Drenckhahn et al., 2008*; *Vincentz et al., 2017*). We confirmed that *Nkx2.5-Cre/Igf2* mutants had normal heart morphology, heart size, and cardiomyocyte size both at birth and at P21 (*Figure 2A–F*). Because cardiomyocyte polyploidy is a prominent feature of postnatal heart biology and one that has a significant impact on proliferative and regenerative competence, we measured the percentage of ventricular mononuclear cardiomyocytes and the nuclear ploidy of the mononuclear cardiomyocyte subpopulation in C57BL/6J x A/J F1 strain background-matched control (*Nkx2.5-Cre* only) and *Nkx2.5-Cre/Igf2* mutants (*Figure 2G,H*). We conducted this analysis at P3, which is just prior to the onset of the peak of DNA synthesis that results in polyploidy, and at P7, which is near (but not at) the end of the wave of polyploidization. At P3, control and *Igf2* mutant

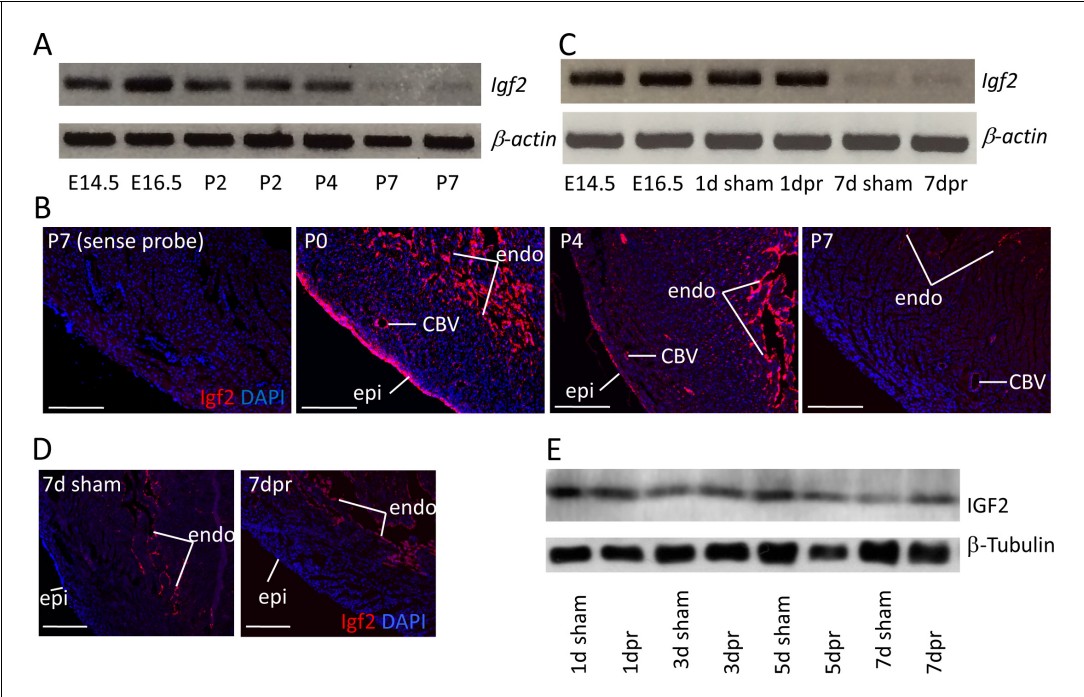

**Figure 1.** *Igf2* expression. (**A**) RT-PCR analysis of *Igf2* mRNA level in whole ventricle tissue at embryonic (E) and postnatal (P) stages, using beta-actin as a reference for sample quality and quantity. (**B**) *Igf2* expression in uninjured neonatal heart ventricle visualized by in situ hybridization (red signal). Expression is detected in epicardium (epi), endocardium (endo), and endothelium of coronary blood vessels (CBV). The first panel (sense probe) indicates the level of tissue background. (**C**) RT-PCR analysis showing that P1 apex resection injury does not change *Igf2* expression at the RNA level at 1 day post resection (dpr) or at 7dpr, relative to sham operated mice. (**D**) In situ hybridization analysis showing that P1 apex resection injury does not change the spatial pattern of *Igf2* expression at 7dpr. (**E**) Western blot of whole ventricle protein from sham-operated and resected hearts; quantitation of three independent blots (*Figure 1—figure supplement 1B*) shows no difference in IGF2 expression. Scale bars in panels B and D: 100μ.
The online version of this article includes the following figure supplement(s) for figure 1:

**Figure supplement 1.** Neonatal IGF2 expression.

hearts were comparable in both mononuclear cardiomyocyte percentage and mononuclear cardiomyocyte nuclear ploidy, with calculated mononuclear diploid cardiomyocyte levels of 80.1 ± 6.4% (*Cre* only) and 85.1 ± 3.5% (*Nkx2.5-Cre/Igf2*). By P7, this had dropped dramatically in the hearts of both genotypes (to 14.7 ± 1.8% and 11.9 ± 1.1%, respectively). We measured a slight difference at P7 in the mononuclear cardiomyocyte percentage that reached statistical significance, although we suspect this to be more likely explained by incidental factors (e.g., the number of pups per litter and the exact ages of each litter; pups from several litters were combined for this analysis). Images of P3 and P7 cardiomyocytes (*Figure 2I*) revealed the hypertrophic enlargement of cardiomyocytes at P7 in both genotypes, which continued to later times (*Figure 2C,F*). Collectively, we conclude that *Nkx2.5-Cre/Igf2* mutants without injury are normal in postnatal heart biology, with the uncertain possibility of a slight difference in mononuclear cardiomyocyte percentage at P7.

To address the role of *Igf2* in neonatal heart regeneration, we performed apex resection on neonatal pups at P1, and evaluated heart morphology after 3 weeks using Sirius red staining in which collagen-rich scar appears bright red and normal myocardium stains yellow. As in other studies (*Mahmoud et al., 2015*; *Wodsedalek et al., 2019*), we distinguished regeneration failure by the presence of transmural scar at the resection point, a partial phenotype by the presence of interspersed scar with abundant myocardium also present, and full regeneration by the absence of scar (examples shown in *Figure 3* and *Figure 3—figure supplement 1*). Because of experimental variability between animals in the amount of apex tissue resected, some variability in outcome is expected, as has been previously documented (*Bryant et al., 2015*). Control mice, all carrying either *Nkx2.5-Cre* or another *Cre* line described below but all wild-type for *Igf2*, predominantly had no scar or in a small number of cases had interspersed scar (*Figure 3*, *Figure 3—figure supplement*

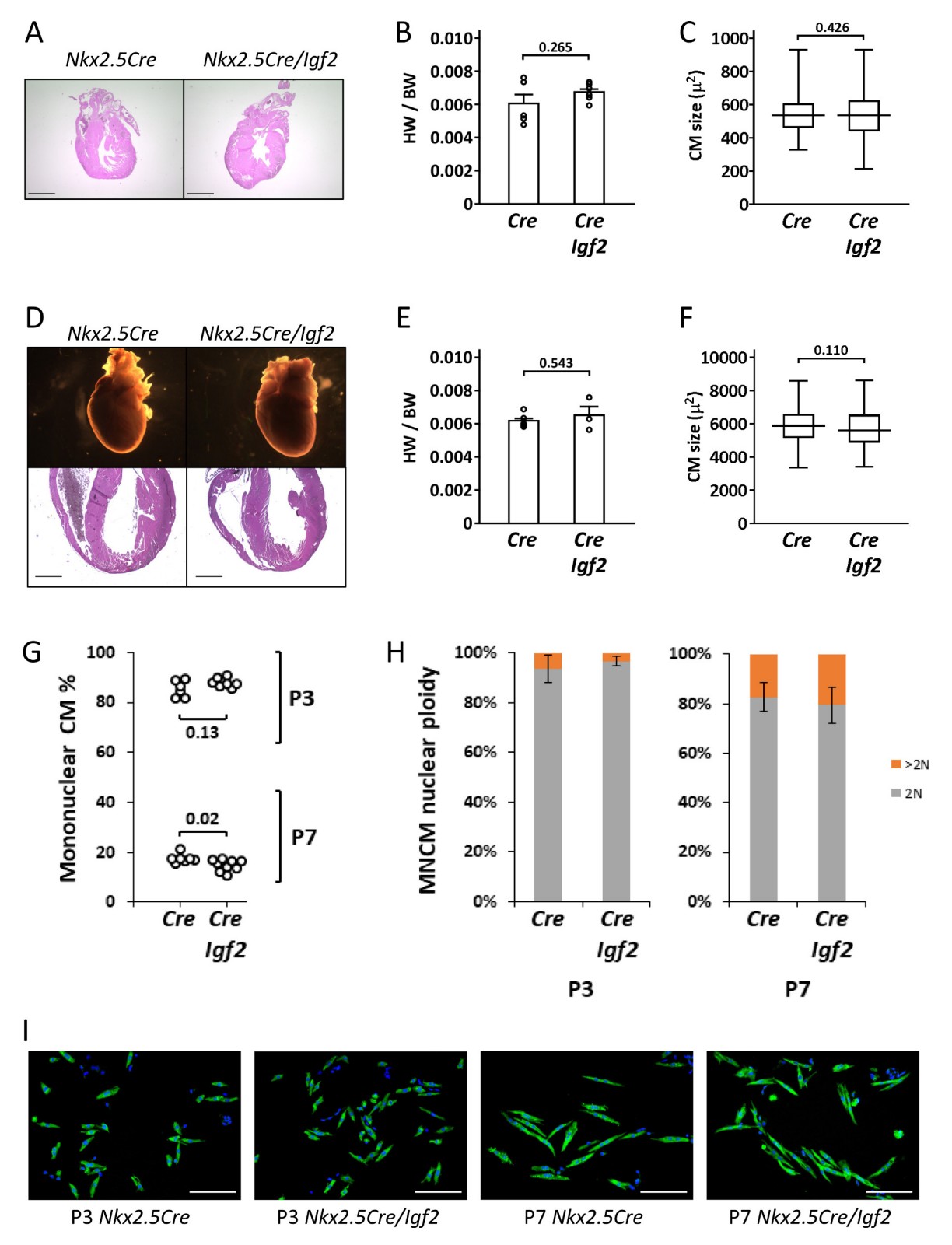

**Figure 2.** Normal parameters of cardiomyocyte biology in postnatal *Nkx2.5-Cre/Igf2* mutants. (A-C), at P1; D-F, at P21. (A,D) Histology showing normal heart morphology. Scale bars: 1 mm. (B,E) Calculation of heart weight to body weight ratio. Differences are not statistically significant. (C,F) Ventricular cardiomyocyte area shown as a box and whiskers plot (median, middle quartiles, and full range of the data indicated; no outliers excluded), based on n = 110 (*Nkx2.5-Cre* P1), n = 110 (*Nkx2.5-Cre/Igf2* P1); n = 115 (*Nkx2.5-Cre* P21), n = 116 (*Nkx2.5-Cre/Igf2* P21) cells. Differences are not statistically

*Figure 2 continued on next page*

*Figure 2 continued*

significant. (G,H) Calculation of ventricular mononuclear cardiomyocyte frequency (G) and the nuclear ploidy of the mononuclear cardiomyocyte subpopulation (H) in cellular preparations from neonates at P3 and P7 as indicated. All pups were on an identical B6AF1 genetic background. The measured difference in mononuclear cardiomyocyte percentage at P7 is statistically significant although of uncertain biological relevance (see text). Differences in nuclear ploidy are not statistically significant (at P3, p=0.19; at P7, p=0.38). (I) Images of cell preparations from the same analysis shown in panels G-H, stained for the cardiomyocyte marker TNNC (green) and DAPI to visualize nuclei; scale bar for all panels = 100μ.

1B, *Table 1*). In contrast, all *Nkx2.5-Cre/Igf2* mutants had transmural scar (*Figure 3*, *Figure 3—figure supplement 1A*, *Table 1*). The presence of scar per se demonstrates only that there is excess and abnormal deposition of matrix. The absence of restoration of the ventricular myocardium, coupled with observations described below of a cardiomyocyte-specific proliferative response, indicate that this phenotype is the manifestation of regeneration failure. Thus, under these typical experimental circumstances, IGF2 signaling is required for neonatal heart regeneration. Furthermore, because of the cardiac mesoderm-restricted recombination domain of *Nkx2.5-Cre*, the source of IGF2 is within rather than from outside the heart.

To define the tissue specific requirements for *Igf2* function during neonatal heart regeneration, we also analyzed conditional *Igf2* mutants using a number of additional *Cre* lines. *Tek-Cre* (also commonly called *Tie2-Cre*) is active in all endothelium, including endocardium, and *Nfatc1-Cre* is active in endocardium and coronary endothelium. We previously showed that *Tek-Cre/Igf2* mutants had no embryonic heart phenotype (*Shen et al., 2015*), and *Tek-Cre/Igf2* and *Nfatc1-Cre/Igf2* mutant mice are both seemingly normal through both prenatal and postnatal life. Collectively, the majority of resected neonates with *Igf2* conditionally deleted in the endocardium and coronary endothelium completely (10/14) or partially (2/14) failed to regenerate (*Figure 3*, *Table 1*). In contrast, the majority (6/8) of resected neonates with *Igf2* conditionally deleted in the epicardium using *Tbx18-Cre* were fully restored 3 weeks later. These results strongly implicate the endocardium and coronary endothelium as the primary if not exclusive lineage source of IGF2 needed for neonatal heart regeneration. This is particularly surprising because the sole source of IGF2 in the embryonic heart that supports normal embryonic cardiomyocyte proliferation is the epicardium, with no apparent embryonic role whatsoever for *Igf2* expressed in the endocardial/endothelial lineage (*Shen et al., 2015*).

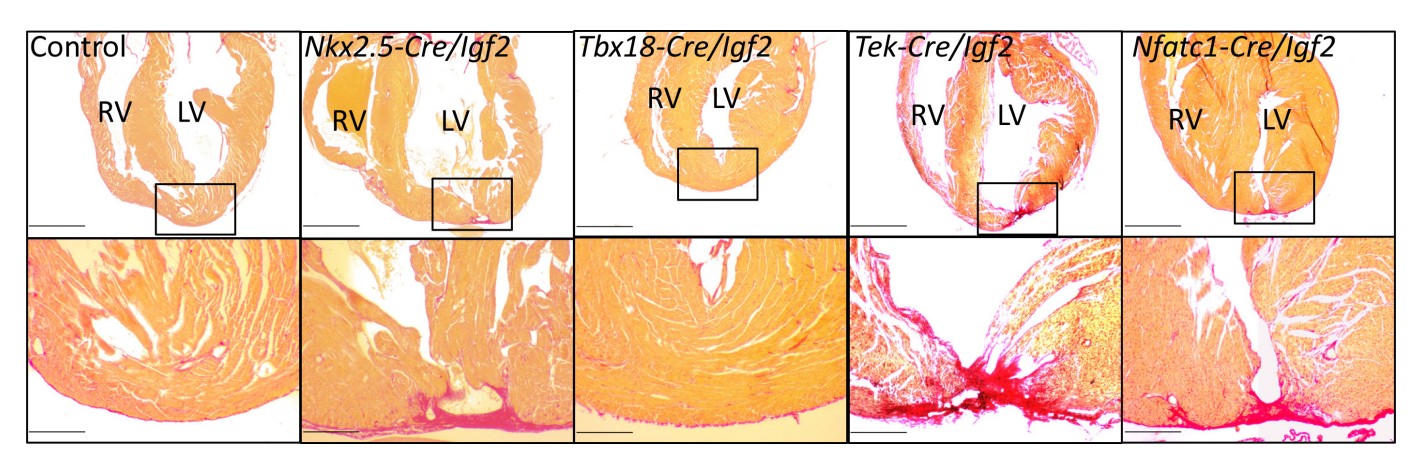

**Figure 3.** Representative examples of successful or failed regeneration at P21 after P1 apex resection in mice of different genotypes. For each heart, serial sections were taken and stained with Sirius red to visualize collagen, and the section with the most extensive degree of staining at the injury site was used to define the degree of regeneration. Additional examples are shown in *Figure 3—figure supplement 1*, and all outcomes are compiled in *Table 1*. The boxed region in each upper panel is shown at higher magnification below. Scale bars: upper, 1 mm; lower, 200μ.
The online version of this article includes the following figure supplement(s) for figure 3:

**Figure supplement 1.** Additional examples of P21 hearts after P1 apex resection, to accompany *Figure 3* and *Table 1*.

**Table 1.** Regeneration after P1 heart apex resection in neonates of various genotypes.
Hearts were evaluated 21 days after resection. Examples of each class of outcome are shown in the figures. Control neonates carried one of the *Cre* alleles listed in the table but were wild-type for the *Igf2* gene. Some *Nfatc1-Cre/Igf2* mutants in this table were nontransgenic littermate controls of the *Nfatc1-Cre/Igf2/mCAT* neonates in Figure 6B.

| Genotype | Total | No scar | Interspersed scar | Transmural scar |
|---|---|---|---|---|
| Control | 23 | 20 | 3 | 0 |
| *Nkx2.5-Cre/Igf2* | 5 | 0 | 0 | 5 |
| *Tek-Cre/Igf2* | 7 | 2 | 1 | 4 |
| *Nfatc1-Cre/Igf2* | 7 | 0 | 1 | 6 |
| *Tbx18-Cre/Igf2* | 8 | 6 | 0 | 2 |
| *Myh6-Cre/Insr* | 7 | 0 | 1 | 6 |
| *Myh6-Cre/Igf1r* | 5 | 4 | 0 | 1 |

## Insulin receptor mediates IGF2 action in neonatal heart regeneration

IGF2 can signal through the insulin receptor (INSR) and through the IGF1 receptor (IGF1R) (*Scalia et al., 2001*). In the embryonic myocardium, only IGF1R is required to mediate IGF2 signaling, despite the seemingly comparable expression of the INSR (*Wang et al., 2019*). In testing the function of these receptors in neonatal heart regeneration, we unexpectedly found that only the insulin receptor is required; there was little if any impact of *Igf1r* deficiency alone (*Table 1*, *Figure 4A*). The use of cardiomyocyte-specific *Myh6-Cre* in this analysis also demonstrates that INSR function in cardiomyocytes mediates IGF2 response.

INSR and IGF1R are both members of the receptor tyrosine kinase family, and are autophosphorylated in response to ligand as the first step in initiating intracellular signaling. By immunofluorescence using antibodies specific for phospho-INSR and for phospho-IGF1R, at 3 and 7 days following sham surgery, there was no basal level of phosphorylation of either receptor, indicating that the activity of the IGF/insulin signaling axis in the normal uninjured neonatal heart is below detection using this method (*Figure 4B,C*; *Figure 4—figure supplements 1A* and *2*). IGF2 protein is present in the heart during this period (*Figure 1E*; *Figure 1—figure supplement 1B*), but seemingly in a manner that does not activate a substantial level of signaling (see below and Discussion). Resection caused prominent INSR phosphorylation (*Figure 4B*, *Figure 4—figure supplements 1A* and *2*), in a pattern that was somewhat heterogeneous even among nearby cardiomyocytes (*Figure 4—figure supplement 1B*); the explanation for this heterogeneity is unknown. INSR phosphorylation was prominent in cardiomyocytes adjacent to the resection plane as well as distal to the injury (*Figure 4B*, *Figure 4—figure supplements 1C* and *2*), which is consistent with the observation that cardiomyocyte proliferation is activated throughout the myocardium after neonatal injury (*Porrello et al., 2011*). Importantly, INSR phosphorylation was eliminated in *Nkx2.5-Cre/Igf2* conditional ligand mutants (*Figure 4B*, *Figure 4—figure supplement 2*), confirming that IGF2 is the ligand that activates INSR during neonatal heart regeneration. Because IGF2 expression at the RNA and protein level is not changed by injury (*Figure 1*; *Figure 1—figure supplement 1B*), we assume that injury releases the block that prevents signaling in the uninjured heart. We observed no IGF1R phosphorylation in injured (or uninjured) neonatal hearts (*Figure 4C*, *Figure 4—figure supplement 1A*), which correlates with the absence of genetic requirement for this receptor in regeneration (*Table 1*). Absence of IGF1R phosphorylation in injured conditional *Igf2* mutants also demonstrates that IGF1 (which signals only through the IGF1R) does not compensate for loss of IGF2. As a positive control for reagent quality and specificity, we detected IGF1R phosphorylation but no activation of the INSR in embryonic heart (*Figure 4—figure supplement 3A*), consistent with our previous observations (*Wang et al., 2019*).

To confirm these results, we performed Western blotting (*Figure 4D*) using ventricular lysates from 3dpr hearts from either control (*Nkx2.5-Cre* only) or from mutants in which the *Insr* gene was conditionally deleted by *Nkx2.5-Cre*. In this case, a phospho-specific antibody that equally

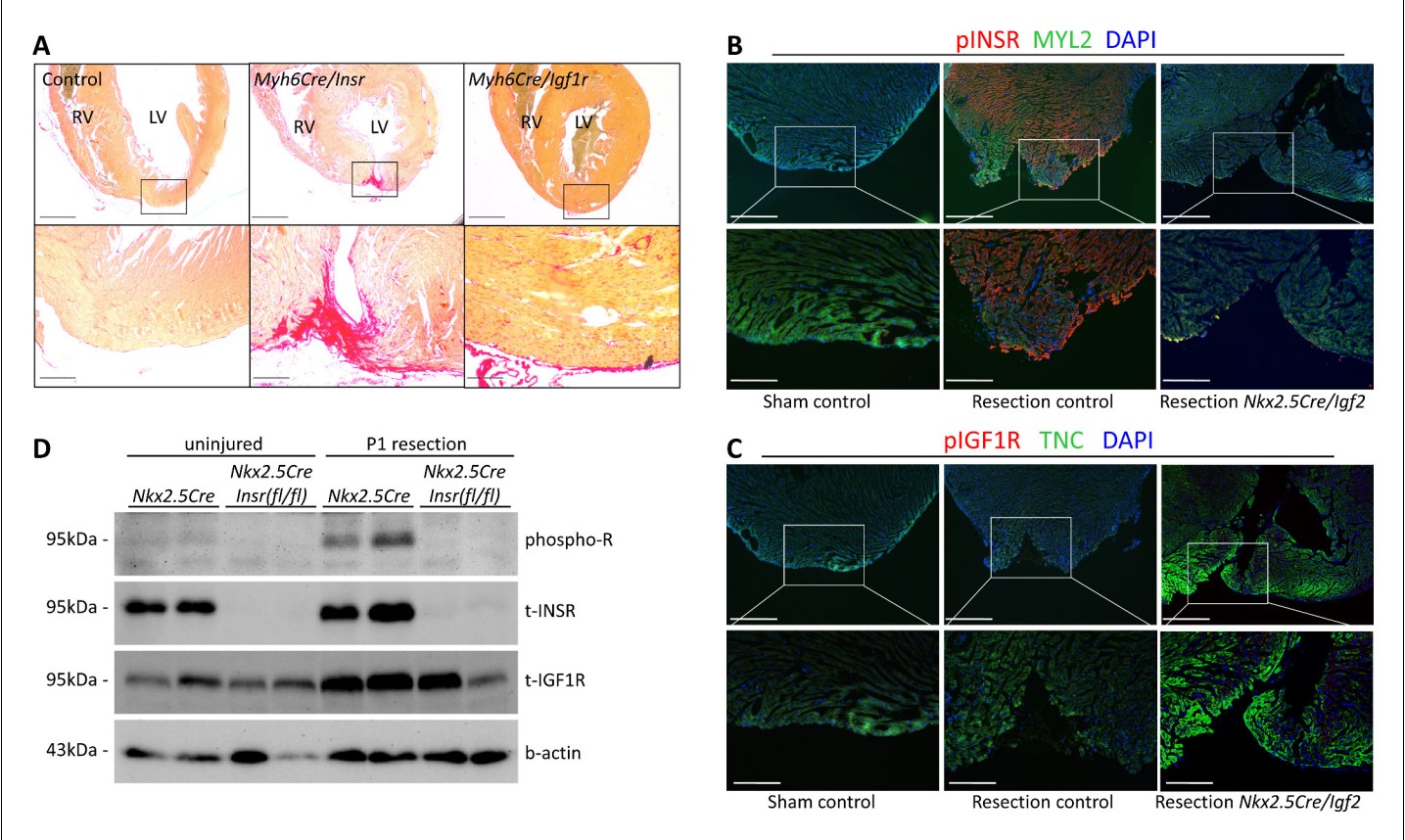

**Figure 4.** INSR is required for neonatal heart regeneration. (**A**) Sirius red staining of P21 hearts after P1 resection, as in **Figure 3**. (**B**) Immunofluorescence detection of phosphorylated INSR at 7dpr is eliminated when *Igf2* is conditionally mutated; there is no signal in the absence of injury (sham). (**C**) No detectable phosphorylation of IGF1R regardless of injury or *Igf2* status in the 7dpr neonatal heart. Cardiomyocytes in B and C were labeled with antibodies for MYL2 (ventricular myosin light chain) or TNC (troponin C), respectively. See similar results at 3dpr for both INSR and IGF1R in **Figure 4—figure supplements 1** and **2**. A positive control for pIGF1R antibody quality is shown in **Figure 4—figure supplement 3**. (**D**) Western blot using ventricular lysates from P1 uninjured and resected hearts at 3dpr, comparing control to *Insr* conditional mutants. The phospho-specific antibody recognizes both pINSR and pIGF1R and so is labeled here simply as phospho-R. Antibodies uniquely recognizing total INSR and total IGF1R (both are approx. 95 kDa), and b-actin as a loading comparison, are also shown. The initial phospho-R blot was stripped and reprobed with the t-INSR antibody; a separate blot run at the same time using the same amount of sample was used for t-IGF1R and actin.

The online version of this article includes the following figure supplement(s) for figure 4:

**Figure supplement 1.** INSR activation after neonatal heart injury.
**Figure supplement 2.** Confocal imaging of phospho-INSR staining at 3dpr.
**Figure supplement 3.** IGF1R and INSR expression and activation.

recognizes the phosphorylated forms of INSR and IGF1R was used; the comparison of controls to conditional *Insr* mutants allowed us to infer the extent to which INSR vs. IGF1R are phosphorylated basally and in response to injury. We detected a very faint signal of phosphorylated receptor in uninjured control heart tissue that was absent from conditional *Insr* mutants; this indicates a low basal level of INSR phosphorylation even though this was not detected by immunofluorescence (**Figure 4B**). Injury at P1 resulted in a prominent increase in phosphorylated receptor in 3dpr controls that was absent in *Nkx2.5-Cre/Insr* mutants. This analysis confirms that P1 heart injury selectively induces INSR phosphorylation but not IGF1R activation (even though IGF1R protein is present).

The insulin receptor gene is transcribed into two isoforms that differ only by the inclusion or exclusion of a 36 bp alternatively spliced exon 11, resulting in proteins that differ by 12 amino acids in an extracellular domain of the receptor (**Frasca et al., 1999**). The shorter form, called INSR-A, mediates signals by both insulin and IGF2, whereas the longer form, called INSR-B, responds to only insulin (**Frasca et al., 1999**; **Scalia et al., 2001**). Throughout the neonatal period, the ratio of the A

and B isoforms was roughly equal, and was not obviously altered after resection (*Figure 4—figure supplement 3B,C*). Although analyzed here only at the RNA level, it is unlikely that the two transcripts are differentially translated (because they differ only in one internal exon), and the two proteins have a similar stability when expressed in transfected cells (*Pandini et al., 2002*). Thus, the insulin receptor A isoform that mediates IGF2 signals is present during the time when IGF2-INSR signaling is required for neonatal heart regeneration, and as demonstrated above is activated by IGF2 after neonatal heart injury.

## *Igf2* deficiency distinguishes cardiomyocyte cell cycle entry and cardiomyocyte proliferation

Neonatal heart apex resection induces cardiomyocyte proliferation as a key element of the regenerative response (*Porrello et al., 2011*). To better understand the requirement for IGF2 signaling in this process, we performed an analysis (*Figure 5*) using the marker phospho-histone H3 (pH3), which labels the nuclei of cycling cells starting at the early prophase stage of mitosis. We colabeled with NKX2.5, which at the protein level is only present in cardiomyocyte nuclei. Three days after P1 sham surgery, a basal level of pH3+ staining was observed, reflecting the ongoing entry of cardiomyocytes into cell cycle that is followed in most cases by polyploidization (*Soonpaa et al., 1996*). There was no difference in control (*Cre*-only) vs. *Igf2*-deficient hearts in the uninjured 3dpr level of pH3+ cardiomyocytes (*Figure 5B*).

In genetically normal pups, P1 resection caused a substantial increase in the number of pH3+ cardiomyocytes at 3dpr. This response was completely abrogated in conditional *Igf2* mutants (*Figure 5B*), indicating that IGF2 is required for this injury-induced mitogenic activity. There is no indication of any additional activity at 3dpr that promotes pH3+ staining, at least in a manner that is independent of IGF2 (*Figure 5B'*).

In genetic control mice (*Cre*-only), the level of pH3+ staining 7 days after sham surgery was several-fold lower than in sham-operated pups at 3dpr, as most cardiomyocytes have already undergone DNA replication and polyploidization by this time (*Soonpaa et al., 1996*). Absence of *Igf2* did not change the basal level of pH3+ staining in sham-operated pups at 7dpr (*Figure 5C*), just as at 3dpr. Injury at P1 resulted in robust cardiomyocyte pH3+ labeling that was still evident in control neonates at 7dpr, just as at 3dpr. Importantly, though, the percentage of pH3+ cardiomyocytes at 7dpr was also elevated in conditional *Igf2* mutants compared to sham-operated animals (*Figure 5C*). This indicates the presence of an IGF2-independent activity that induces cardiomyocyte cell cycle entry at 7dpr, whereas IGF2 is required for all of the injury-induced pH3+ labeling at 3dpr. Approx. 60% of the injury-induced pH3+ labeling at 7dpr is IGF2-independent (red bar in *Figure 5C'*), and 40% attributable to IGF2 (green bar in *Figure 5C'*). It should be noted that the IGF2-independent activity at 7dpr is not necessarily in response to alternative mitogenic factors, as cardiomyocyte-intrinsic responses (recognition of strain or pressure, etc.) might also initiate cell cycle entry at this time.

Phospho-histone H3 is a marker of cells in early mitosis, and does not necessarily indicate that cytokinesis will ensue. It is well established that injury in the adult heart induces cell cycle entry, including S-phase DNA replication and induction of pH3+ staining, but followed mostly by a failure to complete cytokinesis, resulting in cardiomyocytes with increased ploidy rather than an increased number of cardiomyocytes (*Zebrowski et al., 2016*). For the same reason, injury at P7 in wild-type pups is not followed by enough proliferation to support myocardial restoration, as observed in previous studies (*Porrello et al., 2011*; *Puente et al., 2014*). Our data support an interpretation that neonatal heart regeneration requires cardiomyocyte proliferation, that proliferation occurs during the first few days after P1 resection when a high percentage of cardiomyocytes are still able to divide, and that IGF2 acting as a typical mitogen is required during these first few days to induce cell cycle entry that then culminates in proliferation. In contrast, even though there is abundant cell cycle activity (e.g., pH3+ staining) at later times (e.g., at 7dpr), this is not followed by enough cytokinesis (proliferation) to support myocardial restoration. Presumably, the main outcome of cell cycle activity at P7 is to increase the ploidy level of preexisting cardiomyocytes, rather than to increase the number of cardiomyocytes by proliferation.

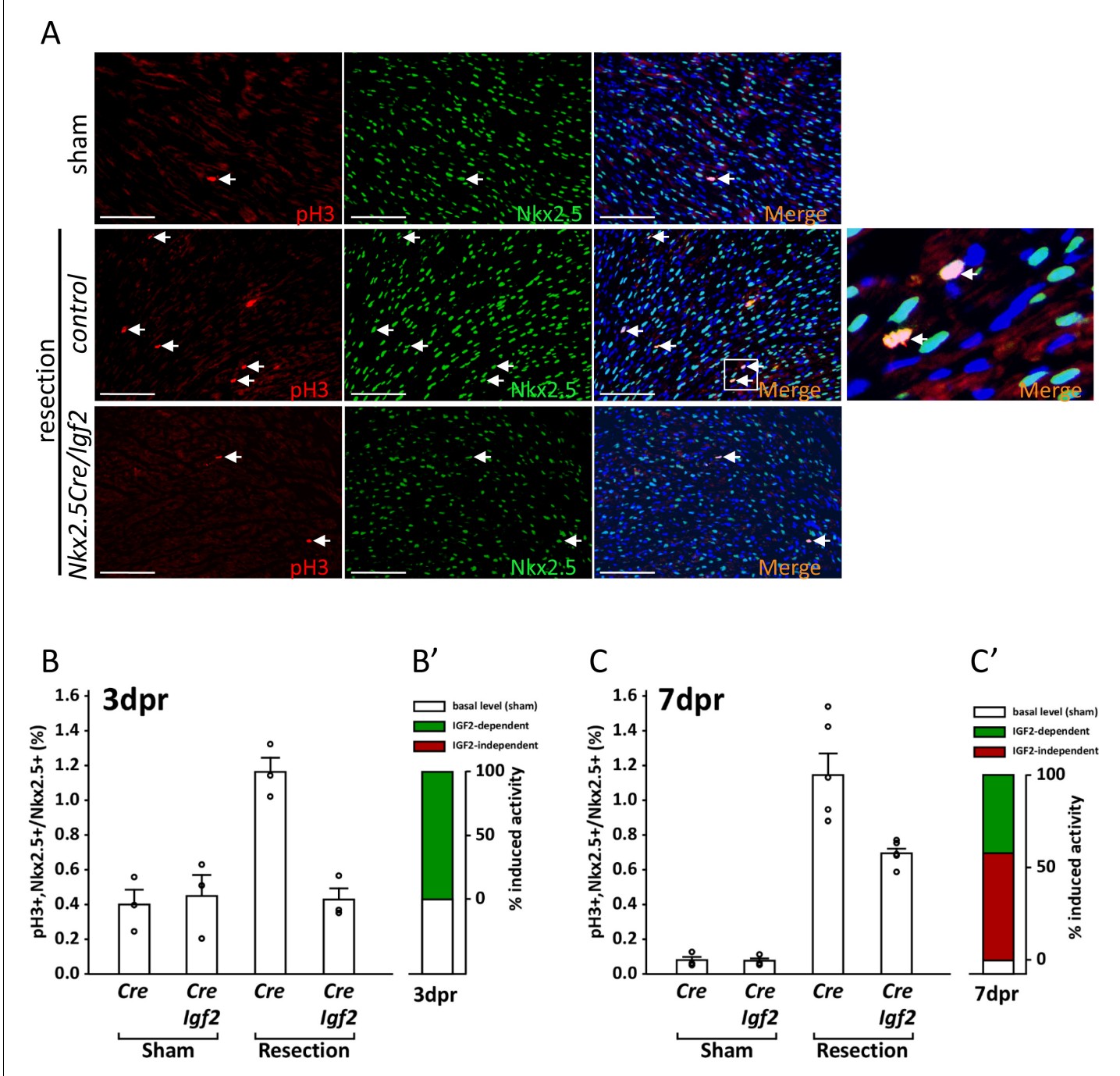

**Figure 5.** Measurement of cell cycle entry activity by phospho-histone H3 (pH3) staining. (A) Representative sections through sham-injured and resected hearts at 3dpr showing visualization of pH3 and NKX2.5 (a cardiomyocyte-specific nuclear marker). Arrows point to double labeled nuclei, one region of which is expanded in the panel at far right. (B) Quantitation of pH3+ cardiomyocyte nuclei at 3dpr. (B') Compilation of the data of panel B, the baseline level of pH3+ cardiomyocytes in sham-operated pups (the average of both genotypes) is in white, and the higher level induced by injury is in green. Because *Igf2* mutation completely eliminates injury-induced pH3+ labeling, there is no indication of IGF2-independent cell cycle entry activity (red). (C) Quantitation of pH3+ cardiomyocyte nuclei at 7dpr. (C') Compilation of the data of panel C, the substantial IGF2-independent injury-induced pH3 labeling activity is indicated by the red bar.

# Delayed cardiomyocyte cell cycle arrest and increased mononuclear diploid cardiomyocyte percentage rescue neonatal heart regeneration in *Igf2* mutants

Several experimental manipulations that lower the level of reactive oxygen were previously shown to prolong the postnatal window during which cardiomyocytes remain proliferative (*Puente et al., 2014*). One method was to raise newborn pups in a mildly hypoxic environment (15% oxygen). We adapted this protocol to address the influence of IGF2 and other activities in heart regeneration in the early and later phases of the first postnatal week. Pregnant females were switched to a 15% oxygen environment just prior to birth, with apex resection of the pups at P1 and evaluation at P21. Regeneration in genetic control mice (n = 9) was unaffected by mild hypoxia (i.e., occurred effectively), just as under normoxic conditions. Impressively, as evaluated by histology, mild hypoxia was sufficient to rescue heart regeneration in *Nkx2.5-Cre/Igf2* mutants: 4/5 displayed no scar and 1/5 displayed interspersed scar 21 days after resection (*Figure 6A*). This contrasts to pups raised in normoxic conditions, where all *Nkx2.5-Cre/Igf2* mutants showed transmural scar (*Table 1*). The amount of scar was quantified to confirm the significant effect of mild hypoxia in mutant hearts (*Figure 6D*; group 2).

Transgenic expression of catalase directed to mitochondria (mCAT) lowers oxidative stress, and like mild hypoxia was previously shown to prolong cardiomyocyte proliferation (*Puente et al., 2014*). Here, we combined the *mCAT* transgene with conditional *Nfatc1-Cre/Igf2* mutation under normoxic conditions, and after P1 injury observed complete regeneration in 3 of 3 pups (*Figure 6B, D*). Rescue of heart regeneration after P1 injury in *Igf2* mutant neonates by mild hypoxia and by expression of the *mCAT* transgene minimizes the likelihood of an explanation for either other than through their common established mechanism to lower oxidative stress and thereby delay the postnatal onset of cardiomyocyte cell cycle arrest. Since the early (P3) IGF2-dependent mitogenic activity is absent in *Igf2* mutants (*Figure 5B*), one explanation for rescue of regeneration is a more effective proliferative response to the later mitogenic activity that is present around P7 even in mutants (*Figure 5C*).

In mice, almost all ventricular cardiomyocytes transition from mononuclear diploid to multinuclear and/or polyploid during the first postnatal week; a small percentage persist as mononuclear and diploid. This transition occurs concurrent with the loss of efficient regenerative ability (*Soonpaa et al., 1996*; *Alkass et al., 2015*). Recent experimental evidence demonstrates that mononuclear diploid cardiomyocytes are proliferative and regenerative in adult mouse and zebrafish (*Patterson et al., 2017*; *González-Rosa et al., 2018*). We recently reported that ubiquitous expression of the *mCAT* transgene elevates the percentage of mononuclear cardiomyocytes and of mononuclear diploid cardiomyocytes in the adult mouse heart, with no effect on nuclear ploidy (*Gan et al., 2019*). Measurements in the uninjured adult heart reflect the interruption or completion of mitosis during the first postnatal week, as there is so little cell cycle activity thereafter. We confirmed that *mCAT* expression resulted in a 2-fold increase in mononuclear cardiomyocyte level at P7 (*Figure 6—figure supplement 1A*) as also seen in the adult.

In our recent analysis (*Patterson et al., 2017*), we demonstrated that strain A/J mice have a particularly high level of mononuclear diploid cardiomyocytes in the adult heart. To relate this to neonatal heart regeneration, we first conducted a time course comparison of the polyploidization process in A/J and C57BL/6J neonates (*Figure 6—figure supplement 1B*). Both strains reached their final adult level shortly after P7, with that of A/J mice being 3-fold higher than that of C57BL/6J mice. This is consistent with an understanding that the process of inducing polyploidy is primarily manifest during the first postnatal week, with genetic differences between strains responsible for determining the percentage of cardiomyocytes that become polyploid or remain mononuclear and diploid (*Patterson et al., 2017*). IGF2 is not relevant to this process, as absence of *Igf2* did not change the percentage of mononuclear diploid cardiomyocytes at P3 and at most only minimally at P7 (*Figure 2*).

We utilized the A/J strain background to further test the influence of mononuclear diploid cardiomyocyte frequency on neonatal heart regeneration. For this purpose, the *Nkx2.5-Cre* and conditional *Igf2* alleles (which are unlinked) were backcrossed for seven generations to the A/J background, which on theoretical grounds results in mice that are 98% homozygous for A/J alleles across the genome other than near the *Nkx2.5-Cre* and *Igf2* loci. Male mice heterozygous for the *Nkx2.5-Cre*

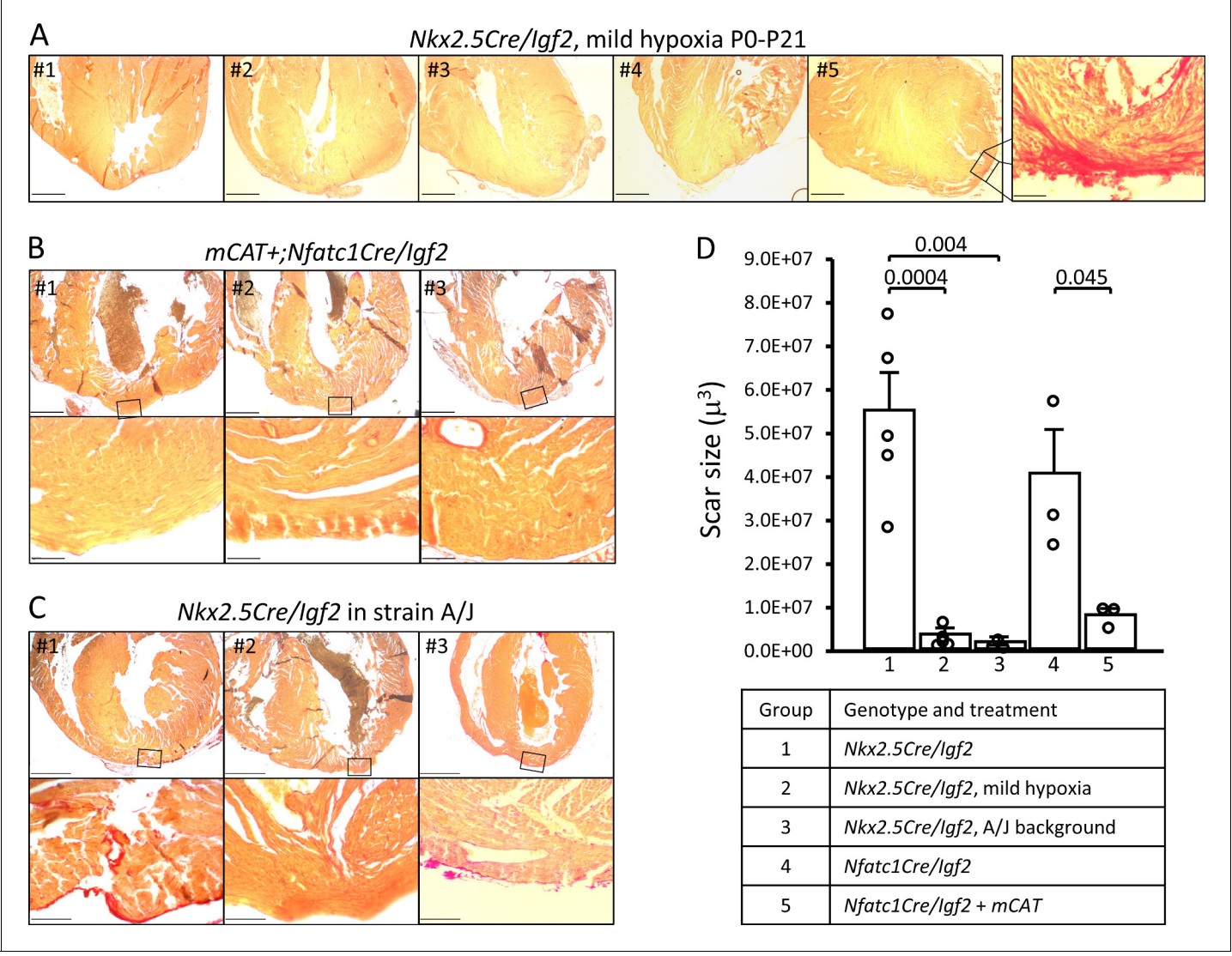

**Figure 6.** Restoration of regeneration in *Igf2* mutant P21 hearts after P1 resection in three conditions that elevate mononuclear diploid cardiomyocyte composition. All analyzed hearts of the indicated genotypes and treatments are shown. As in *Figure 3*, the most extensive degree of Sirius red staining for each heart at the injury site is shown. (A) *Nkx2.5-Cre/Igf2* mutant hearts after mild hypoxia. Heart #5 was scored as having interspersed scar, shown at high magnification in the expanded panel; the other four were scored as having full regeneration. (B) *Nfatc1-Cre/Igf2* mutants also transgenic for the *mCAT* transgene. All three show full regeneration. (C) *Nkx2.5-Cre/Igf2* mutants in the A/J strain background. #1 is classified as interspersed scar, the other two are scored as full regeneration. Scale bars: upper rows, 1 mm; lower rows, 100μ. (D) Scar size was quantitated for *Nkx2.5-Cre/Igf2* and *Nfatc1-Cre/Igf2* mutants and for all of the rescued hearts shown in panels A-C.

The online version of this article includes the following figure supplement(s) for figure 6:

**Figure supplement 1.** Mononuclear cardiomyocyte content.

and conditional *Igf2* alleles on the A/J background were crossed to wild-type female A/J mice, with heart resection performed on neonatal pups at P1 as above. To confirm procedures on this strain background, three genetic control pups (*Cre*-only) were evaluated at 21dpr and all had regenerated, as expected. Importantly, all of 3 *Nkx2.5-Cre/Igf2* pups had also regenerated (*Figure 6C,D*). Thus, even without environmental or transgenic manipulation, when the number of mononuclear diploid cardiomyocytes at P7 is elevated (e.g., here by strain background), the mitogenic (cell cycle entry) activity present around that time (*Figure 5C*) is able to drive a sufficient degree of cardiomyocyte response to accomplish heart regeneration even in P1-injured *Igf2* mutants.

## Myocardial infarction in the adult heart induces *Igf2* reexpression

Unlike the early neonatal heart, in which regenerative capacity is robust, injury to the adult mammalian heart is not followed by substantial cardiomyocyte proliferation and therefore is not effectively regenerated. In the uninjured adult heart, *Igf2* expression was not detectable by in situ hybridization, and was barely detectable by RT-PCR (*Figure 7A,B*). After adult heart injury (permanent coronary artery ligation), we observed that *Igf2* expression was reexpressed in the endocardium in the infarcted zone, strongly at 6 days and continuing although to a declining level at 20 days. This was however not associated with INSR phosphorylation (activation) (*Figure 7C*). This resembles the situation in the uninjured neonatal heart, where *Igf2* is expressed and the protein is present but does not induce receptor activation, and where cardiomyocyte proliferation is minimal.

## Discussion

In the mammalian adult heart, myocardial infarction is typically followed by permanently compromised cardiac function and possible progression to heart failure. This outcome is understood to reflect an extremely limited regenerative capacity of the adult heart. In normal biology, hearts are proliferative during embryonic life and can reactivate proliferation following injury during the initial days following birth, but mostly lose regenerative capacity early in the neonatal period. This study adds several new conceptual elements to understanding the regenerative capacity of the heart.

First, our studies support the view that under normal circumstances, the window of heart regenerative competence in mice is limited to only the first few days of the first postnatal week. A similar temporal conclusion was reached by others (*Ikenishi et al., 2012*; *Notari et al., 2018*), and is a further refinement of the observation that heart injury at P7 is not followed by effective regeneration (*Porrello et al., 2011*). In our study, despite robust cell cycle activity at P7 (*Figure 5C*), *Igf2* mutants failed in heart regeneration after P1 injury. *Igf2* deficiency therefore unmasks early and late first week cell cycle entry activity and distinguishes the necessity of the former for functional tissue regeneration.

Second, IGF2 appears to be the prime if not sole mitogenic factor that directly induces neonatal cardiomyocyte proliferation during the first few days after P1 injury. In *Igf2* mutants there was no indication of any further activity that even partially contributes to an increase in cell cycle entry after P1 injury (*Figure 5B'*). For this reason, effective regeneration does not occur in *Igf2* mutants despite robust cell cycle activity at 7dpr. Other secreted factors previously studied in the context of neonatal heart regeneration (*Kawagishi et al., 2018*; *Singh et al., 2018*; *Das et al., 2019*; *Wodsedalek et al., 2019*) may indirectly influence cardiomyocyte proliferation through alternative cell types or mechanisms, although we cannot formally exclude the possibility that IGF2 and another secreted mitogen are required together to induce proliferation.

Third, it is notable that neonatal heart regeneration relies on paracrine signaling at all. In principle, changes in cardiac output or wall strain after injury might be detected directly by cardiomyocytes, either at myocyte-myocyte junctions or at the cell surface via interaction with extracellular matrix. Such processes (*Bassat et al., 2017*; *Notari et al., 2018*) and others (*Mahmoud et al., 2015*) are clearly relevant but do not substitute for the requirement for IGF2-INSR paracrine signaling. Our conditional manipulation of *Igf2* in cardiac endothelium and cardiomyocyte-specific manipulation of *Insr* establish IGF2 signaling through INSR as the major paracrine pathway that underlies this process.

Fourth, certain circumstances can override the requirement for IGF2 signaling in neonatal regeneration. In this study, we found that mild hypoxia, *mCAT* expression, and an inbred A/J strain background were each individually sufficient to support efficient regeneration in the absence of IGF2. We interpret these outcomes (see model in *Figure 8*) to reflect a higher frequency of completed cell cycle events (i.e., proliferation) among the cardiomyocytes that are induced by P1 injury to enter cell cycle during the later part of the first week, when factors other than IGF2 are present (*Figure 5*). We note a limitation of our analysis is that we have not directly measured proliferation, which for cardiomyocytes is recognized as very difficult to distinguish from polyploidization (*Soonpaa et al., 2013*; *Zebrowski et al., 2016*; *Leone et al., 2018*). Because cardiomyocyte proliferation is necessary for heart regeneration, we infer that rescue of regeneration in *Igf2* mutants must have involved proliferation. Our observations also serve as a caution in interpretation of results when strain background or environment are not properly controlled.

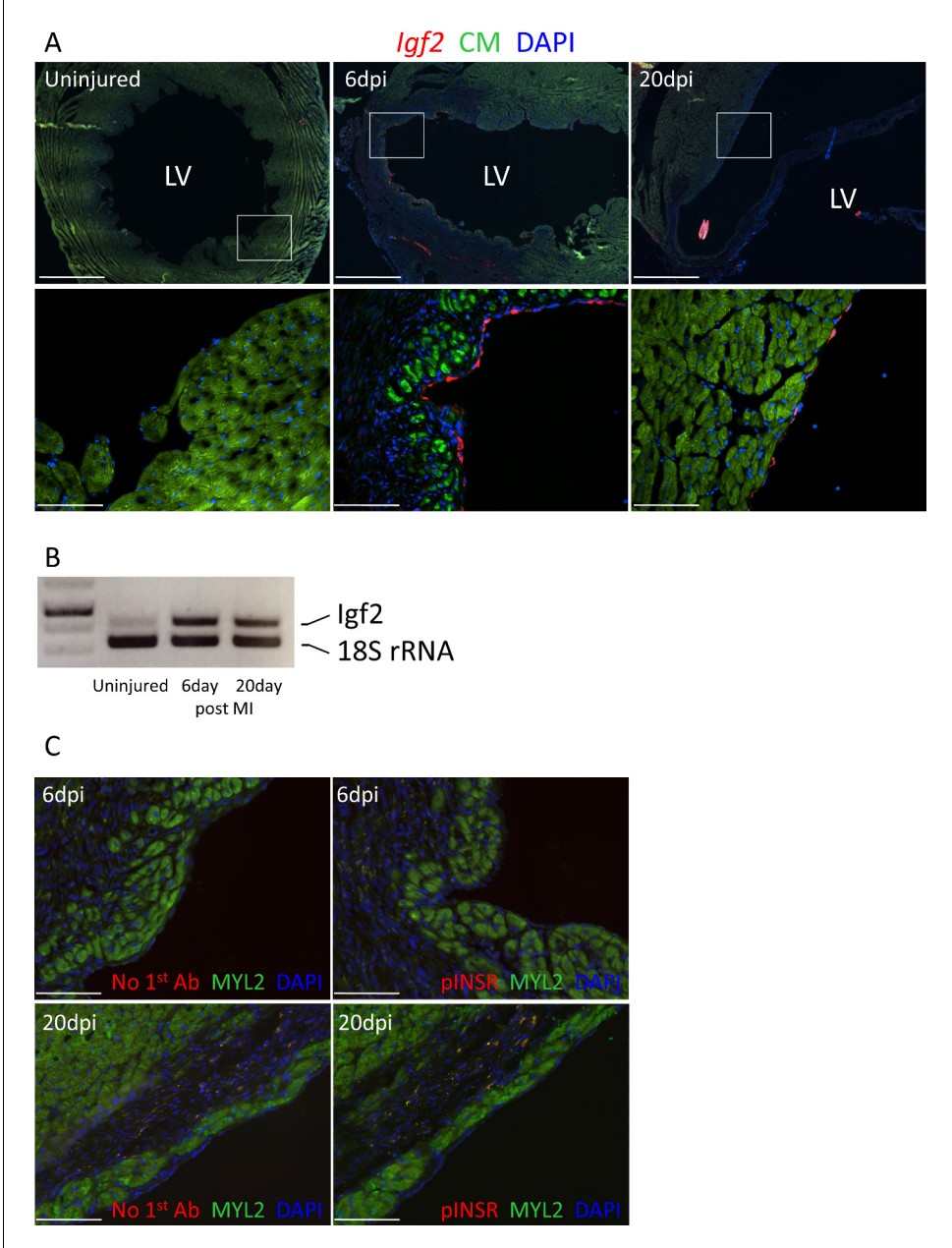

**Figure 7.** Adult heart injury induces *Igf2* expression. (**A**) Sections of adult hearts that were uninjured or at 6 days and 20 days after coronary artery ligation. In the injured hearts, the infarction region is evident by absence of cardiomyocyte autofluorescence (green). *Igf2* is not expressed in the uninjured heart, but is expressed after infarction in the endocardium. (**B**) RT-PCR analysis of *Igf2* expression normalized to 18S rRNA, showing minimal expression in uninjured heart and induction after injury. The first lane is a size marker. (**C**) No activation of INSR phosphorylation after adult heart injury.

Finally, our results implicate the necessity of proliferation-competent mononuclear diploid cardiomyocytes in order to support neonatal heart regeneration. At P1, most cardiomyocytes are mononuclear diploid, and proliferation and regeneration are robust following injury. By P7, under typical circumstances (such as the strain backgrounds and environmental conditions used in most studies), most cardiomyocytes have become polyploid (*Figure 8*), and despite the robust cell cycle entry activity that occurs at this time following P1 injury (*Figure 5C'*), most cardiomyocytes are not able to complete the cell cycle with cytokinesis. By increasing mononuclear diploid cardiomyocyte percentage through P7, regeneration after P1 injury in *Igf2* mutants was rescued, presumably via the non-

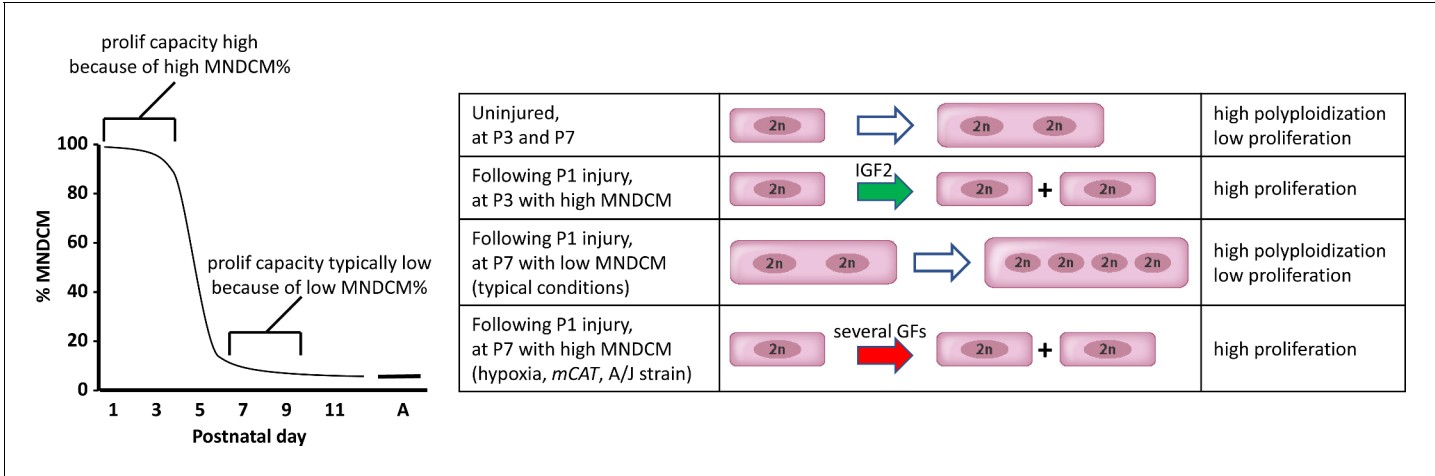

**Figure 8.** A model to explain neonatal heart regeneration. At left is a schematic illustration of the percentage of mononuclear diploid cardiomyocytes (MNDCMs) in the ventricle from the high level at birth to the low level by P7 and later (e.g., as in C57BL/6 mice or in the mixed genetic backgrounds of mice used in most parts of this study; A, adult). IGF2 does not influence how many MNDCMs are present but rather acts on these to promote their proliferation. Consequently, early heart injury is regenerated because the MNDCM percentage is high and can sustain a high level of proliferation in response to IGF2. At right is a representation of the most common type of cell cycle outcome under different circumstances. In the normal postnatal heart in the absence of injury, cell cycle entry leads to polyploidy rather than completed mitosis. In typical hearts (environmentally and genetically normal), P1 injury induces IGF2 expression which induces proliferation at P3 by the high number of MNDCMs, whereas at P7 there are too few MNDCMs present to mount an effective regenerative response even in the presence of other growth factors (GFs) that induce cell cycle activity. With conditions that sustain a higher MNDCM level through and beyond P7, proliferation and regeneration ensue in response to other factors even without prior IGF2 signaling.

IGF2 activity that is evident at 7dpr (*Figure 5C*). The presence of this later activity, which is still not defined, was only possible to detect when the earlier and continuing influence of IGF2 was removed genetically. This interpretation for neonatal heart regeneration is consistent with the importance of cardiomyocyte ploidy in adult heart regeneration (*Patterson et al., 2017*; *González-Rosa et al., 2018*). We note the possibility of two variant interpretations of our results. First, mild hypoxia, *mCAT* expression, and the A/J strain background might influence cardiomyocyte proliferation after neonatal heart injury through other mechanisms, with the increase in mononuclear diploid cardio-myocytes associated with these manipulations being a tangential effect not directly related to regeneration. Second, in addition to serving as a mitogenic signal, IGF2-INSR signaling might also modify the components of cardiomyocyte mitosis to promote proliferation rather than polyploidization. If so, then mild hypoxia, *mCAT* expression, and the A/J strain background must all do the same.

A premise of our interpretations is that the conditional mutant mice used in this study are normal prior to neonatal injury; if this were not true, then their failure to regenerate after injury could reflect a preexisting deficiency rather than an injury-induced requirement for IGF2-INSR signaling. The fact that *Nkx2.5-Cre/Igf2* mutants have a transient embryonic hypoplastic heart phenotype (*Li et al., 2011*) perhaps supports this scenario. We counter this possibility with several observations. First, the embryonic role for *Igf2* is a manifestation of expression in the epicardium, yet epicardium-specific *Tbx18-Cre/Igf2* mutants with the exact same embryonic heart phenotype as *Nkx2.5-Cre/Igf2* mutants (*Shen et al., 2015*) were able to regenerate after P1 injury while *Nkx2.5-Cre/Igf2* mutants were not. Second, both *Tek-Cre/Igf2* and *Nfatc1-Cre/Igf2* mutants have no embryonic heart pheno-type (*Shen et al., 2015*) and yet were unable to regenerate after P1 injury. Third, the INSR has no observable role in heart development (*Wang et al., 2019*) and yet is required for neonatal heart regeneration. These results clearly segregate the consequences of IGF2 signaling in embryonic heart development from its independent role in postnatal heart regeneration. We do note that there is a low level of INSR phosphorylation in uninjured neonatal heart (*Figure 4*), which may have some role or consequence. We evaluated several aspects of postnatal heart biology in uninjured *Nkx2.5-Cre/Igf2* mutants and all were found to be normal (*Figure 2*), with the possible exception of a small dif-ference in the percentage of mononuclear cardiomyocytes at P7 (*Figure 2G*). As commented above, we suspect this to be more likely related to the exact circumstances of the pups that were used for

this analysis (their exact ages and number of pups per litter; both parameters have a prominent impact on polyploidy [*Brodsky et al., 1985*]), rather than an indication of a biologically meaningful difference at P7, although we have not formally disproven the latter. Clearly though, the observation that IGF2 signaling through the INSR is typically required for heart regeneration, and that this requirement can be circumvented by several different manipulations (*Figure 5*) that are unlikely to be relevant to embryonic heart development, supports the interpretation of a unique program of IGF2 signaling that is activated by early postnatal heart injury to promote heart regeneration.

IGF2 is involved in midgestation mouse heart development, so perhaps it is not surprising that it is also involved in neonatal heart regeneration. Nonetheless, with closer examination, there are several features related to its role in neonatal heart regeneration that indicate this to be a unique process and not simply a reiteration of developmental events. First, the lineage sources of IGF2 in heart development and neonatal regeneration are distinct: as shown by genetic manipulation, IGF2 originates from the epicardium (or epicardium-derived mesenchymal cells) in embryonic heart growth, whereas in neonatal heart regeneration, IGF2 derives from the cardiac endothelium (which could include endocardium, coronary endothelium, hematopoietic cells derived from hemogenic endocardium, or endocardium-derived mesenchymal cells). *Igf2* is expressed in both epicardium and endocardium in the embryo and neonate, so the completely opposite lineage source of IGF2 in each context implies unique posttranscriptional modes of signaling regulation. Second, the receptors that mediate IGF2 signaling also differ in the embryo and injured neonate: embryonic development requires IGF1R, whereas neonatal regeneration exclusively uses INSR. The mechanisms that account for this distinction in receptor activity are unknown. Clearly though, when looking at IGF signaling in the embryonic and neonatal heart, the oft-stated paradigm that regeneration is a recapitulation of development is superficially true, but each also has unique features that indicate a deeper complexity and subtlety.

*Igf2* has been studied in the context of zebrafish heart development and adult heart regeneration, via the use of heat shock-regulated ubiquitous expression of an igf1r dominant negative construct and by treatment with an igf1r-specific antagonist (*Choi et al., 2013*; *Huang et al., 2013*). These studies do not provide insight on the lineage sources of IGF2 in the fish heart, but do implicate igf1r as the major receptor in both contexts. At the moment, it is not possible to reconcile the exclusive use of INSR in neonatal mouse heart regeneration with the apparent use of igf1r in zebrafish adult heart regeneration, and it is possible that regeneration in zebrafish has different genetic controls compared to the mouse. Nonetheless, the conservation of reliance on IGF signaling in heart development and regeneration in both species is striking.

It is intriguing that the IGF2-INSR signaling axis is inactive in the uninjured neonatal heart and only becomes active following injury. *Igf2* expression at the RNA and protein levels, and *Insr* expression at the RNA level, are all unchanged by injury, suggesting a posttranslational block to signaling in the normal heart and an alleviation of this block following injury. It is currently unknown what accounts for this block; possible mechanisms could include storage of IGF2 protein in vesicles prior to exocytosis from their source, sequestration after exocytosis by IGF binding proteins or by other components of the extracellular matrix, or some inhibitory feature in uninjured cardiomyocytes that prevents INSR from responding to IGF2. It is interesting that INSR activation (*Figure 4B*, *Figure 4— figure supplements 1* and *2*), cardiomyocyte cell cycle activation (*Figure 5*) and cardiomyocyte proliferation (*Porrello et al., 2011*) are all induced across the entire ventricle following apex resection; this suggests that the alleviation of IGF2 signaling after injury involves a broadly distributed mechanism rather than one localized to the site of injury. How this occurs is unknown, although because this involves IGF2-INSR, there are now molecular targets through which this process can be explored.

A major reason for interest in the very efficient process of neonatal heart regeneration is its contrast with the very limited regeneration seen in the adult heart after injury. Mouse cardiomyocytes transition from being mostly proliferative to mostly nonproliferative during the first postnatal week, although a small subset of cardiomyocytes remain mononuclear diploid and proliferative. Our analysis indicates that regeneration in the neonate requires both a mitogenic stimulus and an adequate population of proliferation-competent cardiomyocytes. It is notable that although *Igf2* is not basally expressed in the adult mouse heart, it is reexpressed following injury, albeit in an inactive manner (*Figure 7*), and possibly inactive through the same mechanisms that prevent IGF2 signaling in the neonate prior to injury. This suggests the possibility that derepressing IGF2 signaling in the adult

heart might promote adult heart regeneration, at least among the population of proliferation-competent cardiomyocytes that are present.

# Materials and methods

## Key resources table

| Reagent type (species) or resource | Designation | Source or reference | Identifiers | Additional information |
|---|---|---|---|---|
| Strain, strain background *Mus musculus* male and female | C57BL/6J, A/J | JAX | 000664, 000646 | |
| Genetic reagent (mouse) | conditional *Igf1r/Insr* | PMID:18650937 | | Laboratory (noncommercial) source |
| Genetic reagent (mouse) | conditional *Igf2* | PMID:22674894 | | Laboratory (noncommercial) source |
| Genetic reagent (mouse) | *Nkx2.5-Cre* | PMID:11783008 | | Laboratory (noncommercial) source |
| Genetic reagent (mouse) | *Tbx18-Cre* | PMID:18480752 | | Laboratory (noncommercial) source |
| Genetic reagent (mouse) | *Tek-Cre* | PMID:11161575 | | Laboratory (noncommercial) source; also commonly called *Tie2-Cre* |
| Genetic reagent (mouse) | *Nfatc1-Cre* | PMID:11786533 | | Laboratory (noncommercial) source |
| Genetic reagent (mouse) | *Myh6-Cre* | JAX | 011038 | |
| Genetic reagent (mouse) | *mCAT* | JAX | 016197 | |
| Antibody | anti-phospho-IGF1R (rabbit polyclonal) | Santa Cruz | SC101703 | for immunofluorescence (IF) (1:50) |
| Antibody | anti-Troponin C (goat polyclonal) | Abcam | ab30807 | for IF (1:500) |
| Antibody | anti-phospho-Insulin receptor (goat polyclonal) | Santa Cruz | SC25103 | for IF (1:50) |
| Antibody | anti-MYL2 (rabbit polyclonal) | Abcam | ab79935 | for IF (1:200) |
| Antibody | anti-rabbit (donkey polyclonal) | Invitrogen | AlexaFluor 488 and 546 | for IF (1:500) |
| Antibody | anti-goat (donkey polyclonal) | Invitrogen | AlexaFluor 488 and 546 | for IF (1:500) |
| Antibody | anti-pH3 (rabbit polyclonal) | Upstate | 06–570 | for IF (1:500) |
| Antibody | anti-NKX2.5 (goat polyclonal) | Santa Cruz | SC8697 | for IF (1:200) |
| Antibody | anti-IGF2 (goat polyclonal) | R&D Systems | AF792 | for Westerns (1:1000) |
| Antibody | anti-phospho-IGF1R/INSR (rabbit polyclonal) | Invitrogen | 700393 | for Westerns; 1.5 µg/ml final concentration |
| Antibody | anti-total INSR (rabbit monoclonal) | Cell Signaling | 3025S | for Westerns (1:1000) |
| Antibody | anti-total-IGF1R (rabbit polyclonal) | Cell Signaling | 3027S | for Westerns (1:1000) |
| Antibody | anti-beta-actin (mouse monoclonal) | Santa Cruz | SC47778 | for Westerns (1:1000) |

*Continued on next page*

*Continued*

| Reagent type (species) or resource | Designation | Source or reference | Identifiers | Additional information |
|---|---|---|---|---|
| Antibody | anti-TnT (troponin T) (mouse monoclonal) | Abcam | ab8295 | for cell staining (1:1000) |
| Antibody | anti-CD31 (Pecam1) (rat monoclonal) | BD Biosciences | 553370 | for cell staining (1:1000) |
| Sequence-based reagent | Insr-F | This paper | PCR primer | GGTGTACTGGGAGAGGCAAG |
| Sequence-based reagent | Insr-R | This paper | PCR primer | CGGTACCCAGTGAAGTGTCT |
| Sequence-based reagent | Igf2-F | This paper | PCR primer | GGCCTTCGCCTTGTGCTGCATC |
| Sequence-based reagent | Igf2-R | This paper | PCR primer | GGATCCACGATCAGGGGACGATGACG |

## Statistical analysis

Primary data are provided in each figure. An unpaired, two-tailed Student t-test, or ANOVA in the case of multiple comparisons, was used to assess statistical significance. By nature of the studies involved, a sample size was not predefined. The number of biological replicate neonates assessed was based on the (random) distribution of genotypes in litters; no technical replicates were included in this study. No outliers in any experimental evaluation were excluded from analysis. ISH and immunostaining assays were conducted on at least six sections per sample and at least three samples per condition (genotype, time, etc.).

## Mice

All animal aspects of this study adhered to the NIH Guide for the Care and Use of Laboratory Animals, and were overseen by the IACUC committee of the University of Southern California. Conditional *Igf1r/Insr* (*Stachelscheid et al., 2008*), conditional *Igf2* (*Haley et al., 2012*), *Nkx2.5-Cre* (*Moses et al., 2001*), *Tbx18-Cre* (*Cai et al., 2008*), *Tek-Cre* (*Kisanuki et al., 2001*), *Nfatc1-Cre* (*Zhou et al., 2002*), and *Myh6-Cre* (*Agah et al., 1997*) alleles used in this study were also used in our prior work (*Li et al., 2011*; *Cavallero et al., 2015*; *Shen et al., 2015*). For most experiments, alleles were maintained on a mixed and unspecified strain background. For convenience, mutant backgrounds are designated simply by the name of the *Cre* line and the conditional gene separated by a diagonal slash. Because the *Igf2* gene is imprinted, conditional *Igf2* mutants were always heterozygous with the conditional *Igf2* allele paternally inherited; all other mutant backgrounds were homozygous. Male mice heterozygous for both *Nkx2.5-Cre* and conditional *Igf2* alleles were mated for seven generations to A/J females (purchased from The Jackson Laboratory, stock number 000646) for studies involving the A/J strain background. *Myh6-Cre* and the *mCAT* transgenic line were purchased from The Jackson Laboratory (stock numbers 011038 and 016197, respectively) and were maintained on a C57BL/6J background unless crossed with alleles carried by mice on outbred backgrounds. Cardiomyocyte ploidy in A/J, C57BL/6J, and C57BL/6J-congenic *mCAT* transgenic mice was evaluated in neonatal mice maintained on these inbred backgrounds. Cardiomyocyte nucleation and nuclear ploidy status in P7 *Nkx2.5-Cre* vs *Nkx2.5-Cre/Igf2* neonates was performed on F1 mice derived by mating A/J-inbred *Nkx2.5-Cre/Igf2* males to wild-type C57BL/6J females.

## Heart injury

The day when newborn pups were observed was defined as P0. The following day, P1 neonatal mice were anesthetized on a bed of ice for 3.5–5 min. After blunt dissection at the left fourth intercostal space, the heart was exposed by gentle squeezing on the body wall. The heart apex was then amputated using iridectomy scissors; sham-operated mice had their hearts exposed without amputation. After surgery, the heart was returned to the body cavity by gentle elevation of the body wall, which was then sutured with 7–0 prolene suture and the skin wound closed with tissue adhesive. Neonates were resuscitated on a warming water bed. In most studies, all neonates of a litter were resected, and genotypes were determined retrospectively at the time of analysis as there is no visible

distinction for any of the genotypes studied in this project. In studies comparing sham vs. resected hearts, an equal number of pups within a litter were randomly chosen for either procedure, sham-operated pups were marked by snipping a small piece of tail, and genotypes were determined retrospectively at the time of analysis. For hypoxia treatment, pregnant females were put into a 15% oxygen chamber (BioSpherix) one day before delivery. P1 apical resection was performed in room air (normal oxygen), and pups were returned to the hypoxic chamber after recovery. Adult mice were injured by permanent left anterior descending coronary artery ligation exactly following procedures described previously (*Patterson et al., 2017*).

## In situ hybridization

Digoxygenin-labelled probes were made as described previously (*Shen et al., 2015*). Heart samples were fixed with 4% paraformaldehyde in PBS at 4°C overnight. After cryoprotection in 10% and 30% sucrose, hearts were embedded and frozen in OCT, and then cryosectioned frontally at 8μ thickness. Sections were fixed in 4%PFA in PBS, washed with PBS, and treated with proteinase K and then triethanolamine in acetic anhydride. Hybridization was performed at 65°C for at least 16 hr. Unhybridized probe was removed by RNaseA digestion. Signal was detected by POD-coupled anti-DIG primary antibody (Roche) and TSAplus Fluorescent Substrate Kit (PerkinElmer).

## Histological studies

Heart samples were dehydrated through increasing ethanol concentrations and then embedded in paraffin. 10μ sections were used for histological studies. To visualize collagen, Picro Sirius red staining (Sigma-Aldrich 365548) or Masson trichrome stain kit (Richard-Allen Scientific) were used. Evaluation of scar at the P21 time of evaluation as transmural, interspersed, or no scar was made from serial sections of each heart; the section with the most extensive degree of Picro Sirius red staining at the injury site was used to define the degree of regeneration. Quantitation of scar was determined by summing the area of scar using ImageJ over complete serial sections through the injury area. Investigators were blinded to genotype at the time of surgery but not at the time of evaluation of extent of scar.

## Immunofluorescence

Heart samples were fixed with 4% paraformaldehyde in PBS at 4°C overnight. After cryoprotection in 10% and 30% sucrose, hearts were embedded and frozen in OCT. 8μ sections were used for IF. Slides were briefly fixed with 4% PFA, then washed and permeabilized with PBS containing 0.1% Triton X-100 at room temperature for three times, 5 min each. After blocking with 10% normal donkey serum and 1% BSA at room temperature for 1 hr, 0.03% Sudan black for 30 min at room temperature was used to block cardiomyocyte autofluorescence. Thereafter, sections were incubated in rabbit anti-phospho-IGF1R (Santa Cruz SC101703; 1:50) and goat anti-Troponin C (Abcam ab30807; 1:500), or goat anti-phospho-Insulin receptor (Santa Cruz SC25103; 1:50) and rabbit anti-MYL2 (Abcam ab79935; 1:200) in PBS containing 1% BSA and 10% donkey serum at 4°C overnight. The secondary antibodies (all used at 1:500) donkey anti-rabbit (Invitrogen Alexa Fluor 546) and donkey anti-goat (Invitrogen Alexa Fluor 488), or donkey anti-goat (Invitrogen Alexa Fluor 546) and donkey anti-rabbit (Invitrogen Alexa Fluor 488) were used to detect pIGF1R and Troponin C, or pINSR and MYL2 with 1 hr incubation at room temperature. For detecting proliferation, primary antibodies used were anti-pH3 (Upstate 06–570; 1:500) and anti-NKX2.5 (Santa Cruz, SC8697; 1:200). Secondary antibodies (1:500) were donkey anti-goat (Invitrogen Alexa Fluor 647) and donkey anti-rabbit (Invitrogen Alexa Fluor 546). Slides were mounted with mounting media containing DAPI. Fluorescent and confocal (1μ optical sections) images were captured at the same settings and at the same time for all sections of an experimental group.

## Polymerase chain reaction

RNA was extracted from heart ventricle using Quick-RNA Mini Prep Kit (Zymo Research). Equal amounts of RNA were used to synthesize cDNA using M-MLV reverse transcriptase (Invitrogen). The following primer sequences were used to amplify Insr cDNA to detect splicing: 5′-GGTGTAC TGGGAGAGGCAAG-3′ and 5′-CGGTACCCAGTGAAGTGTCT-3′. For Igf2, primer sequences were:

5'-GGCCTTCGCCTTGTGCTGCATC-3' and 5'-GGATCCACGATCAGGGGACGATGACG-3'. Beta-actin was used as an internal control.

## Western blot

Proteins separated on SDS-PAGE gels were transferred to polyvinylidene difluoride membranes (Bio-rad). Membranes were blocked in PBS-Tween with 5% nonfat dry milk. Antibody against IGF2 was from R&D (AF792) used at 1:1000 dilution. Bound primary antibodies were visualized using secondary antibodies conjugated to horseradish peroxidase (1:2000, Santa Cruz Biotechnology) and chemi-luminescent substrate (ECL Plus, Thermo Scientific). For quantitation, Western blot images were opened in Image J and using the plot lanes function. All IGF2 bands were normalized to the corresponding bands of beta-tubulin from same membrane. Normalized signal measurement values were analyzed by GraphPad Prism using one-way ANOVA (alpha = 0.05). For evaluation of INSR phosphorylation, *Nkx2.5-Cre* with conditional *Insr* (both alleles heterozygous) adults were mated to homozygous conditional *Insr* partners, and hearts from P4 pups were collected and frozen and then retrospectively genotyped. Western blots prepared as above were probed with anti-phospho-INSR/IGF1R (Invitrogen 700393, 1.5 µg/ml final), anti-INSR (Cell Signaling 3025S (4B8), 1:1000), anti-IGF1R (Cell Signaling 3027S, 1:1000), and beta-actin (Santa Cruz sc-47778, 1:1000), and visualized as above.

## Cardiomyocyte isolation and mononuclear and mononuclear ploidy quantitation

Because strain background strongly influences both nuclear ploidy and mononuclear cardiomyocyte level, mice for this analysis were all on a uniform B6AF1 strain background (described above). P3 and P7 neonatal mouse hearts were rapidly excised after decapitation and gently cleaned to remove blood in ice-cold Kruftbrühe (KB) solution (*Malliaras et al., 2013*). Hearts were mounted to a Langendorff perfusion apparatus, perfused with 37°C oxygenated calcium-free Tyrode's solution at 1 ml/min for 1 min, then digested with 37°C oxygenated calcium-free Tyrode's solution containing 1 mg/ml collagenase type II (Gibco 17101–015) at 1 ml/min for 5 min. After digestion, atria and valves were removed and ventricular tissue alone was excised and triturated in KB solution. The resulting cell suspension was filtered through a 250 µm mesh, stained with LiveDead Fixable (ThermoFisher, L10120) for 20 min at room temperature, then fixed in 2% PFA at room temperature for 15 min. Fixed cell suspensions were stained for cTnT (1:1,000, Abcam ab8295) overnight at 4°C followed by goat anti-mouse secondary (1:500, ThermoFisher A11001) and DAPI. Cell suspensions were pipetted across a slide and coverslipped. Cardiomyocyte nucleation was quantified on an Olympus BX41 fluorescence microscope with a 20x objective. Only live cardiomyocytes were counted; at least 300 cells were counted per heart. For nuclear ploidy analysis, cell preparations prepared as above were also stained with anti-CD31 antibody (BD Pharmingen 553370; 1:250) to label endothelial cells and photographed. Using ImageJ software, nuclei were identified and outlined with a standard threshold requirement, and DAPI fluorescence intensity of individual nuclei in mononuclear cardiomyocytes and in endothelial cells was calculated. The median value of DAPI fluorescence intensity of CD31+ endothelial cell nuclei was used as a diploid nucleus standard, and cardiomyocyte nuclei were defined as being diploid if their normalized intensity values were within a 0.5–1.5x range of this standard. Investigators were blinded to genotype at the onset of and through the completion of this analysis.

## Acknowledgements

This project was supported by NIH grant HL070123 to HMS, and by American Heart Association grant 17SDG33400141 and NSF (MADE in SC) OIA-1655740 to GT. PG was supported by a predoctoral fellowship from the American Heart Association.

# Additional information

## Funding

| Funder | Grant reference number | Author |
|---|---|---|
| National Institutes of Health | HL070123 | Henry Sucov |
| American Heart Association | 17SDG33400141 | Ge Tao |
| National Science Foundation | OIA-1655740 | Ge Tao |
| American Heart Association | | Peiheng Gan |

The funders had no role in study design, data collection and interpretation, or the decision to submit the work for publication.

## Author contributions

Hua Shen, Conceptualization, Data curation, Formal analysis, Investigation, Methodology; Peiheng Gan, Investigation; Kristy Wang, Formal analysis, Investigation, Methodology; Ali Darehzereshki, Methodology; Kai Wang, Investigation, Methodology; S Ram Kumar, Conceptualization; Ching-Ling Lien, Conceptualization, Supervision; Michaela Patterson, Conceptualization, Formal analysis, Investigation, Methodology; Ge Tao, Conceptualization, Formal analysis, Methodology, Writing - review and editing; Henry M Sucov, Conceptualization, Supervision, Funding acquisition, Project administration

## Author ORCIDs

Ching-Ling Lien (iD) http://orcid.org/0000-0002-5100-9780
Michaela Patterson (iD) http://orcid.org/0000-0002-3805-4181
Henry M Sucov (iD) https://orcid.org/0000-0002-3792-3795

## Ethics

Animal experimentation: This study was performed in strict accordance with the recommendations in the Guide for the Care and Use of Laboratory Animals of the National Institutes of Health. All of the animals were handled according to approved institutional animal care and use committee (IACUC) protocol 10173 of the University of Southern California and protocols 2018-00642 and 2018-00310 of the Medical University of South Carolina.

## Decision letter and Author response

Decision letter https://doi.org/10.7554/eLife.53071.sa1
Author response https://doi.org/10.7554/eLife.53071.sa2

# Additional files

## Supplementary files

• Transparent reporting form

## Data availability

All data generated or analysed during this study are included in the manuscript and supporting files.

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
