## [Decision Letter]

Thank you for submitting your article "Mononuclear diploid cardiomyocytes support neonatal mouse heart regeneration in response to paracrine IGF2 signaling" for consideration by *eLife*. Your article has been reviewed by three peer reviewers, one of whom is a member of our Board of Reviewing Editors, and the evaluation has been overseen by Edward Morrisey as the Senior Editor.

The reviewers have discussed the reviews and the Reviewing Editor has drafted this decision to help you prepare a revised submission.

The reviewers agree that the results from the manuscript are interesting and potentially support an important regulatory mechanism in postnatal cardiac regenerative capacity. However, there were several concerns involving experimental design, rigor, and interpretation that must be addressed prior to further consideration at *eLife*. Essential revisions are provided here.

Essential revisions:

1) The major events of postnatal heart composition and growth must be evaluated in *Nkx2.5Cre*/IGF2 mutants. While the developmental abnormalities are rescued, the composition of the heart prior to injury has not been rigorously assessed. In short, the reviewers were not convinced that hearts from *Nkx2.5Cre*/IGF2 postnatal animals were normal prior to injury. To address this concern, *Nkx2.5Cre*/IGF2 mutants must be compared to controls as follows: (1) percentages of mono and binucleated over the first week of postnatal life should be quantified with representative images of mono and binucleated cells shown; (2) nuclear ploidy in mono and binucleated cells should be quantified and compared between cohorts; (3) timing of cell cycle exit should be determined (can be extrapolated from data in (1); (4) hypertrophic growth should be assessed and compared.

2) A more detailed analysis of the rescued animals must be performed. In particular, sample sizes must be increased, scar size must be quantified, and all bullet points from reviewer 1 must be assessed. The rigor of data in Figure 6—figure supplement 1 should be improved (reviewer 1, bullet point 3; reviewer 3, comment 7).

3) The model should be refined in a revision. Specifically, it is unclear how more MNDCs arise in the rescued animals when IGF2 (the sole mitogen at P3) is missing. If the authors find that the percentage of MNDCs is statistically increased prior to injury in *Nkx2.5Cre*/IGF2 mutants under mild hypoxic conditions/mCat+ tg/A/J strain, then where do the authors think that the extra MNDCs are coming from? The authors should include a paragraph in the Discussion section with a much more explicit model that is consistent with their data.

4) Immunostainings must be improved. These data should be supported with Western blots and quantification that is subjected to statistical tests of significance.

5) Statistical tests should be improved by increasing sample sizes.

Reviewer #1:

In the manuscript, "Mononuclear diploid cardiomyocytes support neonatal heart regeneration in response to paracrine IGF2 signaling", Shen et al. demonstrate that IGF2 is required in the endocardium to support cardiomyocyte proliferation during neonatal heart regeneration by signaling through the INSR receptor. They show that in the absence of IGF2 signaling, cardiomyocyte proliferation falls to

basal levels, suggesting that IGF2 is the sole mitogen at P3. At P7, IGF2 is responsible for ~40% of cardiomyocyte proliferation while ~60% is IGF2-independent. They then go to show that increasing the percentage of MNDCs through different experimental mechanisms rescues the IGF2-mediated

regeneration failure. While the data are of high quality and the text is easy to follow, there are concerns with the rescue data and how MNDC numbers influence IGF2-dependent cardiomyocyte proliferation.

1) It is inferred that higher percentages of MNDCs is the reason that the 3 experimental conditions rescues heart regeneration in the absence of *Igf2*. A more detailed analysis of these recused animals should be performed for this argument to be convincing:

• CM proliferation should be evaluated at P7 in all three types of rescued *Igf2* mutants. This should be performed and reported.

• The percentage of MNDCs should be quantified in the 3 experimental situations.

• It appears that only mononucleation was evaluated (Figure 6—figure supplement 1) in the absence of DNA content in only some of the experimental conditions. Since the message is focused on MNDCs, ploidy should also be determined.

• The conclusion that *Igf2* mutants have no difference in MNDC percentage is not convincing. The spread seems very large compared to mCAT+ and A/J-C57BL/6J.

2) Where do the authors think that the additional MNDCs come from in the *Igf2* mutant hearts under mild hypoxic conditions, in the presence of the mCAT transgene, or in the A/J background? There appears to be an internal inconsistency – if *Igf2* is the sole mitogen that supports MNDC proliferation, then how can there be additional MNDCs in its absence? While there is some loose speculation provided in the Discussion, a more refined model should be presented with increased clarity.

Reviewer #2:

In mice the regenerative capacity is lost by seven days after birth. One contributing factor to the loss of regeneration is that the CMs become postmitotic at P7. The authors investigate the role of IGF2 in cardiac regeneration and find that IGF2 is required for neonatal mouse heart regeneration.

Interestingly, IGF2 originates from the endocardium/endothelium lineage and is transduced in cardiomyocytes by the insulin receptor.

1) The authors perform a careful analysis of the source of IGF2 and conclude that the source is endocardium and endothelium. These data are very convincing and rigorously performed.

2) The authors work out the receptor that mediates signaling to the myocardium. Again this is very convincing.

3) The connection to hypoxia and ploidy is fascinating and adds further impact to this manuscript. These are important observations for the field of tissue regeneration.

Reviewer #3:

The study by Shen et al. uses multiple mouse genetic approaches to examine the requirements for *Igf2* in postnatal cardiac regeneration after injury. These studies built on previous reports from the same group that loss of *Igf2* causes defects in myocardial growth during development, but that mice recover to develop normally. These same mice are used in the current study to examine the requirements for *Igf2* in cardiac regeneration after injury. Additional conditional mutants are used as evidence for a paracrine IGF2 signal in postnatal cardiac regeneration.

The results are interesting and potentially support an important regulatory mechanism in postnatal cardiac regenerative capacity. However, there are several major concerns with the experimental design, rigor of data presentation, and interpretation that weaken the major conclusions of the study.

1) The authors conclude that IGF2 expressed in endothelial cells is signaling to the cardiomyocytes. However, it is not clear where IGF2 is lost using the *Nkx2.5Cre* mediated approach. This Cre driver is expressed in multiple tissues near the heart during embryogenesis and loss of IGF2 in proepicardial cells or pharyngeal structures could affect the alterations in cardiac regeneration observed. In addition there are no data showing specific loss of IGF2 expression in the endothelial cells. There are other Cre lines such as Cdh5CreER or Tcf21CreER that would be more appropriate for these studies. As presented in the current manuscript, the major conclusion of the study is not well supported.

2) The *Nkx2.5Cre/Igf2* mutants have developmental abnormalities in cardiac growth that are recovered by birth. It seems important to know if the major events of cardiomyocyte cell cycle arrest, binucleation, and transition to hypertrophic growth are normal in these animals in the postnatal period (P0-P15) in the absence of injury. If these parameters are affected in the mutants, it could alter the interpretation of the results.

3) In general, major findings are not quantified or subjected to statistical tests of significance. Sample sizes are small n=3 in most cases which would not be sufficient to determine significance of the results. This is especially a concern with conclusions regarding cardiac regenerative capacity in the different mutants. Scar sizes need to be quantified and compared to appropriate controls. "Regenerative score" is not really quantitative and individual data points are not shown.

4) Antibody stainings for pINSR and pIGF1R are not entirely convincing. MYL2 and TNC (troponin?) stainings also seem to be variable. These images should be improved and supported by Western blots with quantification and statistical tests of significance.

5) Western blots and qPCR data need to be quantified and subjected to statistical tests.

6) Is there any compensation for loss of *Igf2* by Igf1 which is expressed in the postnatal period? Is circulating *Igf2* affected?

7) Inclusion of images of mono and binucleated cardiomyocytes in Figure 6—figure supplement 1 would be informative. It looks like the *CreIgf2* hearts may be different from controls. Were binucleated cardiomyocyte percentages different?

8) Statistical tests used are not appropriate in many cases because the sample sizes are too small to determine parametric or nonparametric distribution.

---

## [Author Response]

Essential revisions:1) The major events of postnatal heart composition and growth must be evaluated in Nkx2.5Cre/IGF2 mutants. While the developmental abnormalities are rescued, the composition of the heart prior to injury has not been rigorously assessed. In short, the reviewers were not convinced that hearts from Nkx2.5Cre/IGF2 postnatal animals were normal prior to injury. To address this concern, Nkx2.5Cre/IGF2 mutants must be compared to controls as follows: (1) percentages of mono and binucleated over the first week of postnatal life should be quantified with representative images of mono and binucleated cells shown; (2) nuclear ploidy in mono and binucleated cells should be quantified and compared between cohorts; (3) timing of cell cycle exit should be determined (can be extrapolated from data in (1); (4) hypertrophic growth should be assessed and compared.

As now shown in Figure 2G-H, we analyzed both mononuclear cardiomyocyte (CM) level and nuclear ploidy in control (*Nkx2.5Cre* only) and *Nkx2.5Cre/Igf2* mutants, at postnatal days P3 and P7. These two time points bracket the P4-P6 period when most CMs become polyploid. It is not practical to conduct this analysis in the middle of this period (e.g., at P5) as the 1 in 4 yield of control and mutant pups necessitates their recovery from at least several litters, and the exact timing and litter sizes of different litters vary too much during this P4-P6 window to perform this evaluation in a consistent manner. Our new data at P3 and P7 show that controls and mutants become polyploid with comparable kinetics. Our numbers indicate a slight difference in the mononuclear CM level at P7 that had statistical significance. We suspect this is not a true difference in *Igf2* mutant hearts but rather a consequence of slightly different timing and litter sizes between the different litters of pups that were used at this time point. Because we had a predefined experimental procedure and endpoint, we elected to report the data as collected rather than obtain additional numbers that might “correct” this minor discrepancy; this is keeping with current standards of appropriate scientific conduct. We discuss this result in the revised text, and we also provide several additional comments and observations (in the Discussion) on why the general concern about heart composition in mutant neonates that was voiced in the decision letter is unlikely to be relevant. Regarding the subject of hypertrophic growth, we had already addressed this feature (see Figures 2C and F), but have added images (as requested by the reviewers) of P3 and P7 cell preparations to Figure 2I that show a large number of mononuclear and binucleated CMs so that readers can further evaluate this feature for themselves.

2) A more detailed analysis of the rescued animals must be performed. In particular, sample sizes must be increased, scar size must be quantified, and all bullet points from reviewer 1 must be assessed. The rigor of data in Figure 6—figure supplement 1 should be improved (reviewer 1, bullet point 3; reviewer 3, comment 7).

We understand the perspective that a larger sample size would provide more assurance for the rescue results. Because of experimental limitations (whole litters can be lost after surgery if the mothers do not nurse the pups, many individual pups die after surgery either from hemorrhage or from lack of nursing, only one in four surviving pups has the appropriate genotype, and the generational time needed to expand the mouse colony to undertake the requested further evaluations), we cannot respond to this suggestion in the time available. However, the reviewer was appropriate in asking for quantitation of scar size, and this analysis (Figure 6D) clearly illustrates the degree of rescue with very high statistical significance even with small sample numbers. Of the four bullet points of reviewer 1, three have been addressed (mononuclear CM percent in A/J and mCAT mice is in Figure 6—figure supplement 1, nuclear ploidy is in Figure 2H, and better evaluation of *Igf2* mutants is in Figure 2G-I). The reviewer also asked for an evaluation of CM proliferation in rescued *Igf2* mutants, although did not specify what type of experimental demonstration would be appropriate. The field recognizes the difficulty in documenting CM proliferation (i.e., completed mitosis), which is distinct from cell cycle entry which we did show (e.g., Figure 5), and to confirm this specific point would require a substantial investment of time. Because CM proliferation is necessary for heart regeneration, we infer that rescue of regeneration in *Igf2* mutants must involve CM proliferation, but agree that this is a limitation of our analysis. We have therefore added discussion of this limitation to the revised text. Finally, the similar question from reviewers 1 and 3 about the data in Figure 6—figure supplement 1 has been addressed by the removal of a previous panel in this supplementary figure and the inclusion of new and more extensive data on the same point now provided in Figure 2G-I.

3) The model should be refined in a revision. Specifically, it is unclear how more MNDCs arise in the rescued animals when IGF2 (the sole mitogen at P3) is missing. If the authors find that the percentage of MNDCs is statistically increased prior to injury in Nkx2.5Cre/IGF2 mutants under mild hypoxic conditions/mCat+ tg/A/J strain, then where do the authors think that the extra MNDCs are coming from? The authors should include a paragraph in the Discussion section with a much more explicit model that is consistent with their data.

This comment was a misunderstanding of our model by the reviewer, which indicates that we did not explain this adequately in the original manuscript. We do not think that IGF2 influences how many mononuclear diploid CMs are present; rather, IGF2 (the only mitogen at P3) and other growth factors and IGF2 (at P7) act on these CMs to promote their proliferation. Under typical circumstances, the number of MNDCMs is too low at P7 to achieve sufficient regeneration, but this number is elevated by each of the three rescue conditions, thereby allowing regeneration via the other factors even when IGF2 is not present. The confusion may have arisen from the model diagram (in Figure 8) that in retrospect was originally drawn in a way that imposed IGF2 on how many MNDCMs are present. That diagram and the overall figure have been revised, and the text has also been revised in a way that we hope makes the model more clear.

4) Immunostainings must be improved. These data should be supported with Western blots and quantification that is subjected to statistical tests of significance.

This comment (extracted from a lengthier point made by reviewer 3) refers to the immunofluorescence data shown in Figure 4 and associated supplementary figures. We were surprised by this comment as we think the primary result – that there is no INSR phosphorylation before injury, that INSR phosphorylation occurs after injury in an IGF2-dependent manner, and that IGF1R is not activated by injury – is unambiguous. The concern perhaps arose because of variation in phospho-INSR signal intensity, although we pointed out that there is variation in response even in adjacent CMs (e.g., Figure 4—figure supplement 1B). Regardless, the reviewer requested that we support the immunofluorescence result with Western blot data, which we have now added (Figure 4D), which clearly confirms the induction of INSR phosphorylation by injury and the lack of IGF1R activation.

5) Statistical tests should be improved by increasing sample sizes.

We have added new data with statistical validation in Figures 2G-H and Figure 6D. As noted above regarding evaluation of additional rescued pups (Essential revision 2), this was not practical to do in the time available, but the statistical evaluation of the pups already analyzed (Figure 6D) confirms a very high level of statistical significance.